# Efficiency of Non-Truthful Auctions in Auto-bidding with Budget Constraints

## ABSTRACT

We study the efficiency of non-truthful auctions for auto-bidders with both return on spend (ROS) and budget constraints. The efficiency of a mechanism is measured by the price of anarchy (PoA), which is the worst case ratio between the liquid welfare of any equilibrium and the optimal (possibly randomized) allocation. Our first main result is that the first-price auction (FPA) is optimal, among deterministic mechanisms, in this setting. Without any assumptions, the PoA of FPA is $n$ which we prove is tight for any deterministic mechanism. However, under a mild assumption that a bidder's value for any query does not exceed their total budget, we show that the PoA is at most 2. This bound is also tight as it matches the optimal PoA without a budget constraint. We next analyze two randomized mechanisms: randomized FPA (rFPA) and "quasi-proportional" FPA. We prove two results that highlight the efficacy of randomization in this setting. First, we show that the PoA of rFPA for two bidders is at most 1.8 without requiring any assumptions. This extends prior work which focused only on an ROS constraint. Second, we show that quasi-proportional FPA has a PoA of 2 for any number of bidders, without any assumptions. Both of these bypass lower bounds in the deterministic setting. Finally, we study the setting where bidders are assumed to bid uniformly. We show that uniform bidding can be detrimental for efficiency in deterministic mechanisms while being beneficial for randomized mechanisms, which is in stark contrast with the settings without budget constraints.

**ACM Reference Format:**
Anonymous Author(s). 2023. Efficiency of Non-Truthful Auctions in Auto-bidding with Budget Constraints. In *Proceedings of THE WEB CONFERENCE (WWW'24)*. ACM, New York, NY, USA, 15 pages. https://doi.org/10.1145/nnnnnnn.nnnnnnn

## 1 INTRODUCTION

Auto-bidding has become increasingly popular in online advertising as it allows advertisers[1] to set high-level goals and constraints, rather than manually submitting bids for each

---

[1] We will often use advertisers and bidders interchangeably in this paper since each auto-bidder acts on behalf of a single agent.

*WWW'24, May 13–17, SINGAPORE*
© 2023 Association for Computing Machinery.
ACM ISBN 978-x-xxxx-xxxx-x/YY/MM...$15.00
https://doi.org/10.1145/nnnnnnn.nnnnnnn

individual keyword. A prototypical example is that the advertiser may set a goal to maximize the total volume of conversions subject to a return-on-spend (ROS) constraint and a total budget constraint. There are other possible targets, e.g. maximizing the number of shown ads, and other constraints, e.g. target cost per acquisition (tCPA). In general, we can model such problems as follows: each advertiser $i$ has a value $v_{i,j}$ for query $j$, and advertiser $i$'s goal is to maximize their total value subject to the constraint that their total value exceeds their total spend. Auto-bidding agents then act on behalf of advertisers to solve this optimization problem. This can save advertisers a significant amount of time and effort, and it can also help them to achieve better results from their advertising campaigns. In this paper, we refer to advertisers or bidders that maximize their value subject to a constraint as value-maximizers.

Prior research has shown that the efficiency of auctions for value-maximizing auto-bidders is different from that of the more traditional utility maximizers that aim to maximize their quasilinear utility [1, 6, 8, 14, 16, 23, 25]. Most prior work focused on auto-bidding with only a ROS constraint, or with both ROS and budget constraints but restricted to the Vickrey-Clarke-Groves (VCG) auction [1, 16]. In this paper, we are interested in the price of anarchy (PoA) [4] of non-truthful[2] auctions, such as the first price auction (FPA) and the randomized first price auction (rFPA), for value-maximizing bidders with ROS and budget constraints.

The analysis of the PoA for FPA with both a ROS and a budget constraint is interesting and challenging in several aspects. First, it turns out that the gap between the optimal deterministic allocation and the optimal randomized allocation is large. For FPA without a budget constraint, an optimal solution is to simply allocate each query to the bidder with the highest value. In particular, there is no benefit in using a randomized, or fractional, allocation. However, in Theorem 3.1, we show that in the setting with both ROS and budget constraints, the gap between the optimal integral allocation and the optimal fractional allocation can be as large as $n$, the number of bidders. Given this, we consider the following two notions of efficiency. First, we define PoA as the worst case ratio between the liquid welfare of any equilibrium and the optimal *randomized* allocation. Second, we define the integral PoA (I-PoA) as the worst case ratio between the liquid welfare of any equilibrium and the optimal *deterministic* allocation. Note that the PoA is always at least as large as the I-PoA. From our earlier observation, it follows that any deterministic auction has a PoA of at least $n$. In the setting with both ROS and budget constraints, we ask and answer the following questions in this paper:

- What is the PoA and integral PoA of FPA?

---

[2] In this paper, as in prior works, an auction is "truthful" if it is a dominant strategy for the a traditional quasi-linear to bid truthfully. By "non-truthful", we refer to auctions that may not be truthful.

- Is there a randomized algorithm that guarantees a constant PoA?
- Are there reasonable assumptions that decrease the gap between the optimal fractional and optimal integral allocation? Are there deterministic auctions with a constant PoA under these assumptions?

Deng et al. [16] showed that the PoA of FPA is 2 with a ROS constraint but the PoA becomes 1 if the bidders are assumed to bid uniformly, i.e. each bidder $i$ has a uniform multiplier $m_i$ and places a bid of $m_i v_{i,j}$ for every query $j$. It is natural to ask whether uniform bidding also improves the equilibria of FPA. More specifically:

- Does the uniform bidding assumption make the PoA and integral PoA smaller for FPA with both ROS and budget constraints?
- Does the uniform bidding assumption make the PoA smaller for rFPA?

## 1.1 Our contributions

We study the efficiency of various non-truthful auctions in the auto-bidding setting with both ROS and budget constraints.

We use $n$ to denote the number of bidders. We use Opt (resp. I-Opt) to denote the optimal liquid welfare achievable by a randomized (resp. deterministic) allocation. Note that in an auto-bidding setting with ROS but not budget constraints, we have Opt = I-Opt since the optimal allocation is to deterministically allocate to the bidder with the highest value. In Theorem 3.1, we show that the gap between Opt and I-Opt is at least $n$. This implies that the PoA of an deterministic mechanism is at least $n$. Complementing this, we show that the PoA for FPA is at most $n$. Thus, the optimal PoA for deterministic mechanisms is exactly $n$. This already shows that new techniques will be required for the budgeted case since prior work compared against a deterministic approach in their proofs.

We first study the efficiency of FPA and summarize the results in Table 1. We prove that FPA is an optimal mechanism in a couple settings when there are both ROS and budget constraints. The previous paragraph established that the PoA of any deterministic auction is at least $n$. We prove that this gap is tight in Theorem 3.3. Interestingly, we show that the PoA is at most 2 under the mild assumption that for any bidder $i$, their value for any query $j$ is at most their budget $B_i$. This PoA bound coincides with the setting with only a ROS constraint.

It turns out that the suggested inefficiency of FPA, by incorporating a budget, is due to using a randomized allocation as a benchmark. If we consider a deterministic benchmark instead, we can show that FPA has no additional loss in efficiency when a budget is introduced. We formalize the results in Table 2. Next, we explore the setting where bidders are assumed to bid uniformly. With only a ROS constraint, previous work has shown that, under the uniform bidding restriction, FPA obtains the optimal allocation [16]. Interestingly, enforcing uniform bidding makes the integral PoA much worse for the setting with budget. In fact, we show that the I-PoA is $n$ (tight). Intuitively, when the bidders are assumed to have a single bid multiplier for all queries, they could be in a situation that when they increase the bid multiplier to win some queries, they win

more than what they could afford and violate their budget. Note that we show a lower bound of PoA in Theorem 3.15, which indicates a lower bound of I-PoA. Similarly, we show an upper bound of I-PoA in Theorem 3.16, which indicates an upper bound of PoA.

Next, we analyze the randomized FPA (rFPA) mechanism proposed by Liaw et al. [23] in the setting with both ROS and budget constraints for two bidders, and show that the PoA upper bound of 1.8 in their paper also apply in this setting. Recall that uniform bidding actually decreases the efficiency for FPA (compared against a deterministic benchmark). However, uniform bidding improves the efficiency for rFPA. The results are summarized in Table 3. The high PoA for uniform bidding in deterministic algorithm is because of the high correlation between all bids from the same bidder, that causes a "taking nothing or breaking the budget" situation. For randomized algorithms like rFPA, even when the bids are highly correlated, the bidders are able to increase bids smoothly to get more fractional value. This avoids the cases that cause high PoA in deterministic auctions.

Finally, we consider a "quasi-proportional" FPA mechanism, which chooses each bidder with a probability proportional to a power of their bid. We show that when the power approaches infinity, the PoA of this auction is at most 2 for auto-bidders with both ROS and budget constraints in Theorem 5.1.

## 1.2 Related works

The literature on auto-bidding has focused on various auction mechanisms, constraints, and extra information. The main difference between auto-bidding and traditional quasi-linear bidding is that auto-bidders are value maximizers, while quasi-linear bidders are profit maximizers. Aggarwal et al. [1] proposed a general framework for auto-bidding with value and budget constraints in multi-slot truthful auctions. They showed that it is a near optimal strategy for auto-bidders to adopt a uniform bidding strategy, and that the optimal PoA is 2. Mehta [25] showed that in the setting with two bidders with only a ROS constraint but not a budget constraint, the PoA is at most 1.89 with a randomized truthful auction. Liaw et al. [23] showed that the PoA can be improved to 1.8 using a non-truthful auction which they called randomized FPA. We adopt their methodology and extend this result to the setting with both ROS and budget constraints in Theorem 4.1. They also showed that when the number of bidders tends to infinity, any auction satisfying some mild assumptions has a PoA of at least 2. Both Liaw et al. [23] and Deng et al. [14] showed that the PoA of FPA with the ROS constraint is 2. We extend this setting to have both the RoS and budget constraints, and show that the extra budget constraint brings the PoA of FPA to $n$ (tight). We also study the integral and fractional PoA of FPA when the bidders are assumed to bid uniformly.

Deng et al. [14] also showed that when there are both value maximizers and utility maximizers, the PoA of FPA is at most $1/0.457$ and that this is tight. There is also a line of work that explores how to utilize machine-learned signals in the auto-bidding setting [6, 12, 14, 16]. In particular, they show that by

**Table 1: Price of Anarchy (PoA) for FPA**

| PoA for FPA | ROS + budget | | ROS | |
|---|---|---|---|---|
| | LB | UB | LB | UB |
| Non-uniform bidding | $n$ (Cor 3.2) | $n$ (Thm 3.3) | 2 [23] | 2 [23] |
| Non-uniform, $v_{i,j} \leq B_i$ | 2 (Cor 3.7) | 2 (Thm 3.8) | 2 [23] | 2 [23] |
| Uniform bidding | $n$ (Thm 3.15) | $n$ (Thm 3.16) | 1 | 1 [16] |

**Table 2: Integral Price of Anarchy (I-PoA) for FPA**

| I-PoA for FPA | ROS + budget | | ROS | |
|---|---|---|---|---|
| | LB | UB | LB | UB |
| Non-uniform bidding | 2 (Cor 3.7) | 2 (Thm 3.6) | 2 [23] | 2 [23] |
| Uniform bidding | $n$ (Thm 3.15) | $n$ (Thm 3.16) | 1 | 1 [16] |

**Table 3: Price of Anarchy (PoA) for rFPA with two bidders**

| PoA for rFPA | ROS + budget | | ROS | |
|---|---|---|---|---|
| | LB | UB | LB | UB |
| Non-uniform bidding | – | 1.8 (Thm 4.1) | – | 1.8 [23] |
| Uniform bidding | – | 1.5 (Thm 4.2) | 1 | 1 [16] |

using such signals as reserves and boosts can lead to improved efficiency in the presence of auto-bidders.

The majority of the research mentioned above assumes that there is only a ROS constraint, without any budget constraint. The only exceptions are Aggarwal et al. [1] and Deng et al. [16], which both studied the integral PoA in VCG auctions, where uniform bidding is nearly optimal. In this paper, we focus on a less studied setting: both the integral and fractional PoA of value maximizing auto-bidders with ROS and budget constraints for a number of non-truthful auctions: FPA, rFPA, and quasi-proportional FPA. There is a thread of research on budget pacing in online or repeated auctions for auto-bidders [10, 11, 20, 21, 24] to minimize regret. In this paper, we focus on the price of anarchy of single-shot auctions, which is fundamentally different from budget pacing in online settings in several aspects. First, in the online setting, the bidders follow a specific budget pacing bidding strategy and show that when bidders follow such a strategy, the welfare is at least half of the optimal [24]. In this paper, some of our results are in the setting where the bidders are allowed to bid arbitrarily. Thus, the aforementioned result may not apply in this setting. Second, the online setting assumes that the values are drawn i.i.d. in every round whereas we assume the value may be adversarially chosen. Third, we study the liquid welfare in pure Nash equilibria, where no bidder would deviate from their current bid. In the online setting, people study small regret or diminishing regret bidding strategies when the number of steps approaches infinity, but this is different from a pure Nash equilibrium in the offline setting.

There are also recent related work by Balseiro et al. [5] and Golrezaei et al. [22] which show that both total revenue and welfare can be improved by choosing appropriate reserve prices. Deng et al. [15] studied the PoA of VCG auctions with user cost. Deng et al. [12] studied fairness in auto-bidding with machine learned advice. Ni et al. [28] studied ad auction design with coupons in the auto-bidding world. Other works on auto-bidding include understanding auto-bidding in multi-channel auctions [2, 13], incentive properties of auto-bidding [3, 26], optimal auction design in the Bayesian setting [6], dynamic auctions for auto-bidders [17, 18], and resource allocation / auction for utility maximizers with budget constraints [7, 9].

## 2 PRELIMINARIES

Let $A$ be a set of $n$ advertisers and $Q$ be a set of queries. Each advertiser $i \in A$ has a total budget $B_i$ and a value $v_{i,j}$ for query $j \in Q$. We let $b_{i,j}$ denote advertiser $i$'s bid on query $j$. A single-slot auction is defined via an allocation function $\pi \colon \mathbb{R}_+^A \to [0,1]^A$, where $\sum_{i \in A} \pi_i(b) \leq 1$ for all $b \in \mathbb{R}_+^A$, and a cost function $c \colon \mathbb{R}_+^A \to \mathbb{R}_+^A$ which denotes the price paid by advertiser $i$ if they are allocated the slot. In particular, the expected price paid by advertiser $i$ is $\pi_i(b) \cdot c_i(b)$.

*Background on Auto-bidding.* Let us assume that all advertisers except $i$ have fixed their bids. Then the decision variables for advertiser $i$ are $\{\pi_{i,j}\}_{j \in Q}$ where $\pi_{i,j}$ is the probability that

advertiser $i$ wins the query. The bid $b_{i,j}$ that advertiser $i$ must place to win with probability $\pi_{i,j}$ is implicitly determined by $\pi_{i,j}$. Finally, we let $c_{i,j}$ denote the cost that advertiser $i$ must pay for query $j$ when they win. Note that $c_{i,j}$ may be a function of $\pi_{i,j}$.

The goal of the auto-bidder is to optimize the advertiser's total value subject to (i) a budget constraint where advertiser $i$'s total expected is no more than $B_i$ and (ii) a target return-on-spend (ROS) constraint where advertiser $i$'s total expected value is no less than their total expected spend [3]. More formally, the auto-bidding agent for advertiser $i$ aims to solve the following optimization problem:

$$\text{maximize:} \sum_{j \in Q} \pi_{i,j} v_{i,j}$$

$$\text{subject to:} \sum_{j \in Q} \pi_{i,j} c_{i,j} \leq B_i \quad \text{(Budget)}$$

$$\sum_{j \in Q} \pi_{i,j} c_{i,j} \leq \sum_{j \in Q} \pi_{i,j} v_{i,j} \quad \text{(ROS)}$$

$$\forall j \in Q, \ \pi_{i,j} \in [0, 1].$$

Let $\Pi = \{\pi \in [0,1]^{A \times Q} : \forall j, \ \sum_{i \in A} \pi_{i,j} \leq 1\}$ be the set of all feasible allocations. Given an allocation $\pi$, we let $\text{LW}(\pi)$ be the liquid welfare which is defined as

$$\text{LW}(\pi) = \sum_{i \in A} \min\{B_i, \sum_{j \in Q} \pi_{i,j} v_{i,j}\}.$$

The notion of liquid welfare was introduced by [19] and measures the "willingness to pay" of an allocation. Let $\text{OPT} = \max_{\pi \in \Pi} \text{LW}((\pi)$ be the value of the optimal randomized allocation that maximizes liquid welfare and let $\pi^* \in \text{argmax}_{\pi \in \Pi} \text{LW}((\pi)$ denote one such optimal allocation.

## 2.1 Deterministic allocations

For deterministic allocations, we assume that there is a single bidder with positive probability of winning the query. In other words, $\pi_{i,j} \in \{0, 1\}$ for all $i \in A$ and $j \in Q$. Let $\Pi^I := \Pi \cap \{0, 1\}^{A \times Q}$ be the set of feasible deterministic allocations. Let $\text{I-OPT} = \max_{\pi \in \Pi^I} \text{LW}(\pi)$ be the value of the optimal deterministic allocation that maximizes liquid welfare and let $\pi^{I*} \in \text{argmax}_{\pi \in \Pi^I} \text{LW}(\pi)$ denote one such optimal allocation.

## 2.2 Equilibrium

We say that the bids $\{b_{i,j}\}$ are in an equilibrium if the two statements below holds for each bidder $i$:

(1) Advertiser $i$ satisfies both their ROS and budget constraints: $\sum_{j \in Q} \pi_{i,j} c_{i,j} \leq \min\{B_i, \sum_{j \in Q} \pi_{i,j} v_{i,j}\}$.
(2) Let $\pi$ and $c$ be the resulting allocation and costs of $\{b_{i,j}\}$. Suppose bidder $i$ deviates to bids $\{b'_{i,j}\}_{j \in Q}$, while other bidders remain their bids in $\{b_{i,j}\}$. Let $\pi'$ and $c'$ denote the allocation and costs after bidder $i$'s deviation. Then either bidder $i$ does not gain more value, or bidder $i$

---

[3]One can think of the ROS constraint as an "ex-ante Individual Rationality" constraint that ensures that the advertiser does not pay more than their value, on average.

violates their constraint. Formally, at least one of the following two inequalities is true:

- $\sum_{j \in Q} \pi'_{i,j} v_{i,j} \leq \sum_{j \in Q} \pi_{i,j} v_{i,j}$
- $\sum_{j \in Q} \pi'_{i,j} c_{i,j} > \min\{B_i, \sum_{j \in Q} \pi'_{i,j} v_{i,j}\}$.

When we consider a fixed equilibrium EQ with a deterministic allocation $\pi$, we will use $N(i)$ to denote the set of all queries that bidder $i$ wins in EQ and $O(i)$ to denote the set of all queries that are assigned to bidder $i$ in OPT. For a query $j$, let $\text{SPEND}(j)$ denote the expected spend on query $j$ in the equilibrium. For any bidder $i$, let $\text{SPEND}(i)$ denote the total expected spend of bidder $i$ in EQ. For any subset of bidders $A' \subseteq A$, we write

$$\text{OPT}(A') = \sum_{i \in A'} \min\{B_i, \sum_{j \in Q} \pi^*_{i,j} v_{i,j}\},$$

$$\text{I-OPT}(A') = \sum_{i \in A'} \min\{B_i, \sum_{j \in Q} \pi^{I*}_{i,j} v_{i,j}\},$$

and

$$\text{LW}(A') = \sum_{i \in A'} \min\{B_i, \sum_{j \in Q} \pi_{i,j} v_{i,j}\}.$$

For a single bidder $i$, we abuse the notation and write $\text{LW}(i) = \text{LW}(\{i\})$ and $\text{OPT}(i) = \text{OPT}(\{i\})$.

Given an instance $S$ and a mechanism $\mathcal{M}$, let $\Pi^{\text{EQ}}$ denote the set of allocations with $\mathcal{M}$ at equilibrium. The PoA of $\mathcal{M}$ is defined as $\sup_S \sup_{\pi \in \Pi^{\text{EQ}}} \frac{\text{LW}(\pi^*)}{\text{LW}(\pi)}$. We also define the integral PoA (I-PoA) as $\sup_S \sup_{\pi \in \Pi^{\text{EQ}}} \frac{\text{LW}(\pi^{I*})}{\text{LW}(\pi)}$, which compares the liquid welfare of an equilibrium with the optimal deterministic allocation. In this paper, we study the PoA when a Nash equilibrium exists.

## 3 FIRST PRICE AUCTION

In this section, we study the efficiency of FPA with both ROS and budget constraints. We note that all our conclusions hold for arbitrary deterministic tie-breaking rules. It is straightforward to see that I-OPT = OPT for auto-bidding without a budget constraint: the optimal allocation is to assign each query $j$ to bidder $i^* \in \text{argmax}_{i \in A} v_{i,j}$. However, in the setting with budget constraints, there is a gap between I-OPT and OPT. We first show that I-OPT can be as small as $\frac{1}{n}$OPT. Next, we show that both the PoA and for FPA is at most $n$, and the integral PoA for FPA is at most 2.

THEOREM 3.1. *In auto-bidding with both ROS and budget constraints, there exists an instance with $n$ bidders that* I-OPT $\leq \frac{1}{n}$OPT.

PROOF. We construct the instance as follows. There are $n$ bidders and a single query. Each bidder has a budget $B_i = 1$ and value $v_{i,1} = n$ for the query. The optimal allocation is to assign the query to each bidder $i$ with $\pi_{i,1} = \frac{1}{n}$. This results in a liquid welfare of $n$. The optimal *integral* allocation is to assign the query to a single arbitrary bidder. This has a liquid welfare of 1. $\square$

Since the liquid welfare of any equilibrium of FPA is at most I-OPT, we have the following corollary.

**Corollary 3.2.** *The PoA for FPA is at least $n$ when the bidders have both budget and ROS constraints.*

Remember our conclusions in this section hold for arbitrary deterministic tie-breaking rules for FPA. Note that we can adjust the example in Theorem 3.1's proof to change bidder 1's value and budget to $v_{1,1} = n + \varepsilon$ and $B_1 = 1 + \varepsilon$, where $\varepsilon \to 0$. Consider a case that every bidder bids $b_{i,1} = v_{i,1}$. It is easy to see this is an Equilibrium with deterministic or randomized tie-breaking rules. Therefore, the PoA is still at least $n$ for FPA also with randomized tie-breaking rules.

Next, we show that the PoA of FPA is at most $n$, i.e. the lower bound example is tight.

**Theorem 3.3.** *The PoA for FPA with both ROS and budget constraints is at most $n$.*

**Proof.** We split the bidders into two sets:

$$A_B = \{i \in A \mid B_i \leq \sum_{j \in N(i)} v_{i,j}\} \text{ and } A_{\bar{B}} = A \setminus A_B.$$

Note that, by the definition of liquid welfare, for $i \in A_B$, we have $\mathrm{LW}(i) = B_i \geq \mathrm{Opt}(i)$. Thus, we have

$$\mathrm{LW}(A_B) \geq \mathrm{Opt}(A_B). \tag{1}$$

We further split $A_{\bar{B}}$ in to two sets. We define $A_{\bar{B}1}$ as the set of bidders $i$ for which there exists at least one query $j \notin N(i)$ such that if bidder $i$ wins query $j$ then, in addition to the queries they are currently winning, their total value would exceed $B_i$. In other words, if bidder $i$ gets this extra query $j$, then they will achieve their optimal liquid welfare $B_i$. Next, we define $A_{\bar{B}0} = A_{\bar{B}} \setminus A_{\bar{B}1}$. These are the bidders for which we can add any query not in $N(i)$ and their total value would still be at most $B_i$. Formally, the sets are defined as

$$A_{\bar{B}1} = \{i \in A_{\bar{B}} \mid \exists j \notin N(i), s.t. \ v_{i,j} + \sum_{j' \in N(i)} v_{i,j'} \geq B_i\}$$

$$A_{\bar{B}0} = A_{\bar{B}} \setminus A_{\bar{B}1}.$$

We first bound $\mathrm{Opt}(A_{\bar{B}1})$ and $\mathrm{Opt}(A_{\bar{B}0})$ in Lemma 3.4 and Lemma 3.5 below, and then put everything together using the constraint that the total spend in the auction is upper bounded by the total liquid welfare.

**Lemma 3.4.** *We have $\sum_{i \in A_{\bar{B}1}} \sum_{j \notin N(i)} \mathrm{Spend}(j) + \mathrm{LW}(A_{\bar{B}1}) \geq \mathrm{Opt}(A_{\bar{B}1})$.*

**Proof.** Fix any $i \in A_{\bar{B}1}$ and consider a query $j' \notin N(i)$ such that $v_{i,j'} + \sum_{j \in N(i)} v_{i,j} \geq B_i$. Note that such a query $j'$ must exist by definition of $A_{\bar{B}1}$. We claim that $\mathrm{Spend}(j') \geq B_i - \sum_{j \in N(i)} v_{i,j}$ in any equilibrium of FPA. We will prove this by contradiction, so assume $\mathrm{Spend}(j') < B_i - \sum_{j \in N(i)} v_{i,j}$. Then the highest bid on $j'$ is strictly less than $B_i - \sum_{j \in N(i)} v_{i,j}$. Now, observe that if bidder $i$ bids $B_i - \sum_{j \in N(i)} v_{i,j}$ on query $j'$ then bidder $i$ would win query $j'$. Their total value would then be at least $B_i$ and their total spend would be at most $B_i$. Thus their value has increased while both their budget and ROS constraints remain satisfied. This contradicts the assumption that the bidders are in an equilibrium. We conclude that there exists $j'$, such that $\mathrm{Spend}(j') \geq B_i - \sum_{j \in N(i)} v_{i,j}$ for all bidders $i \in A_{\bar{B}1}$. Using the trivial upper bound $\mathrm{Spend}(j') \leq$

$\sum_{j \in O(i) \setminus N(i)} \mathrm{Spend}(j)$ (since $j' \in O(i) \setminus N(i)$), this implies that $B_i \leq \sum_{j \in O(i) \setminus N(i)} \mathrm{Spend}(j) + \sum_{j \in N(i)} v_{i,j}$ for $i \in A_{\bar{B}1}$. We thus have that,

$$\mathrm{Opt}(A_{\bar{B}1}) \leq \sum_{i \in A_{\bar{B}1}} B_i$$

$$\leq \sum_{i \in A_{\bar{B}1}} \sum_{j \notin N(i)} \mathrm{Spend}(j) + \sum_{i \in A_{\bar{B}1}} \sum_{j \in N(i)} v_{i,j}$$

$$\leq \sum_{i \in A_{\bar{B}1}} \sum_{j \notin N(i)} \mathrm{Spend}(j) + \mathrm{LW}(A_{\bar{B}1}),$$

as claimed. Note that for the first line, we used $\mathrm{Opt}(A_{\bar{B}1}) = \sum_{i \in A_{\bar{B}1}} \min\{B_i, \sum_j v_{i,j} \cdot \pi_{i,j}^*\} \leq \sum_{i \in A_{\bar{B}1}} B_i$. □

**Lemma 3.5.** *We have $\sum_{i \in A_{\bar{B}0}} \sum_{j \notin N(i)} \mathrm{Spend}(j) + \mathrm{LW}(A_{\bar{B}0}) \geq \mathrm{Opt}(A_{\bar{B}0})$.*

**Proof.** The proof is similar to the proof of Lemma 3.4. For $i \in A_{\bar{B}0}$, we claim that for each query $j' \notin N(i)$, $\mathrm{Spend}(j')$ is at least $v_{i,j'}$. Again, we prove this by contradiction. Assume there exists $j' \in O(i) - N(i)$, such that $\mathrm{Spend}(j') < v_{i,j'}$. Then bidder $i$ can bid $v_{i,j'}$ to win query $j'$. Note that

$$c_{i,j'} + \sum_{j \in N(i)} c_{i,j} \leq v_{i,j'} + \sum_{j \in N(i)} c_{i,j} < B_i,$$

where the first inequality uses that bidder $i$ bids and pays $v_{i,j'}$ on query $j'$ and that bidder $i$'s ROS constraint was initially satisfied and the second is by definition of $A_{\bar{B}0}$. But this shows that bidder $i$ can improve their value without violating their constraints, which contradicts the assumption that the bidders are in an equilibrium. Thus, it must be $\mathrm{Spend}(j') \geq v_{i,j'}$ for all $i \in A_{\bar{B}0}$ and $j' \notin N(i)$. In particular, $\sum_{j \notin N(i)} \mathrm{Spend}(j) \geq \sum_{j \notin N(i)} v_{i,j}$. Following a similar argument as in Lemma 3.4, we have that

$$\mathrm{Opt}(A_{\bar{B}0}) \leq \sum_{i \in A_{\bar{B}0}} \sum_{j \in Q} v_{i,j}$$

$$= \sum_{i \in A_{\bar{B}0}} \sum_{j \notin N(i)} v_{i,j} + \sum_{i \in A_{\bar{B}0}} \sum_{j \in N(i)} v_{i,j}$$

$$\leq \sum_{i \in A_{\bar{B}0}} \sum_{j \notin N(i)} \mathrm{Spend}(j) + \mathrm{LW}(A_{\bar{B}0}),$$

where the first inequality is because $\mathrm{Opt}(A_{\bar{B}0}) = \sum_{i \in A_{\bar{B}0}} \min\{B_i, \sum_j v_{i,j} \cdot \pi_{i,j}^*\} \leq \sum_{j \in Q} v_{i,j}$. □

We now return to the proof of Theorem 3.3. Let $N^{-1}(j)$ denote the advertiser that wins query $j$. Combining Inequality (1), Lemma 3.4 and Lemma 3.5, we have

$$\mathrm{Opt} = \mathrm{Opt}(A_B) + \mathrm{Opt}(A_{\bar{B}1}) + \mathrm{Opt}(A_{\bar{B}0})$$

$$\leq \mathrm{LW}(A_B) + \sum_{i \in A_{\bar{B}1}} \sum_{j \notin N(i)} \mathrm{Spend}(j) + \mathrm{LW}(A_{\bar{B}1})$$

$$+ \mathrm{LW}(A_{\bar{B}0}) + \sum_{i \in A_{\bar{B}0}} \sum_{j \notin N(i)} \mathrm{Spend}(j)$$

$$= \mathrm{LW}(A_B) + \mathrm{LW}(A_{\bar{B}1}) + \mathrm{LW}(A_{\bar{B}0}) + \sum_{i \in A_{\bar{B}}} \sum_{j \notin N(i)} \mathrm{Spend}(j)$$

$$\leq \mathrm{LW}(A) + \sum_{j \in Q} \sum_{i \in A \setminus N^{-1}(j)} \mathrm{Spend}(j) \tag{2}$$

$$= \text{LW}(A) + (n-1) \cdot \sum_{j \in Q} \text{Spend}(j) \qquad (3)$$

$$\leq n \cdot \text{LW}(A). \qquad (4)$$

In Eq. (2), we swapped the order of the sum. In Eq. (3), we used that $|A \setminus N^{-1}(j)| = n-1$ (each query is assigned to exactly one bidder). In Eq. (4), we used that the liquid welfare is an upper bound on the total spend. □

Although the PoA of FPA is large because of the large gap between OPT and I-OPT, the theorem below shows that the I-PoA of FPA with both ROS and budget constraints is at most 2, which is the same as in the setting with only a ROS constraint.

**THEOREM 3.6.** *The I-PoA for FPA with both ROS and budget constraints is at most 2.*

This proof and all other proofs not included in the main paper are relegated to the appendix.

Corollary 3.4 of [23] shows that the I-PoA of FPA without budget is at least 2. It is straightforward to generalize it to obtain the following corollary, by assuming all the budgets are infinity. Note that the lower bound of I-PoA is also a lower bound of PoA.

**COROLLARY 3.7.** *The I-PoA for FPA with both ROS and budget constraints is at least 2.*

Next, we make a mild assumption that $v_{i,j} \leq B_i$ for all $i \in A$ and $j \in Q$, i.e. any bidder's value for any query is no more than their budget. With this assumption, we are able to show that the PoA is at most 2 for FPA in Theorem 3.8. Indeed, our assumption avoids the "large value" cases represented by the example in Theorem 3.1 to achieve a small PoA bound.

**THEOREM 3.8.** *If $v_{i,j} \leq B_i$ for every $i \in A$ and $j \in Q$ then the PoA of FPA with both ROS and budget constraints is at most 2.*

**PROOF.** We split the bidders into two sets:

$$A_B = \{i \in A \mid B_i \leq \sum_{j \in N(i)} v_{i,j}\} \text{ and } A_{\bar{B}} = A \setminus A_B.$$

By the definition of liquid welfare, for $i \in A_B$, we have $\text{LW}(i) = B_i \geq \text{OPT}(i)$. Thus, we have

$$\text{LW}(A_B) \geq \text{OPT}(A_B). \qquad (5)$$

For $i \in A_{\bar{B}}$, define $\mathcal{D}(i) = \{j \notin N(i) \mid \text{Spend}(j) < v_{i,j}\}$. Let $V_i = \sum_{j \in N(i)} v_{i,j}$ denote the total value that bidder $i$ has in the equilibrium EQ. Note that $V_i = \text{LW}(i)$ for $i \in A_{\bar{B}}$ because $B_i \geq V_i$. For a set $N' \subseteq N(i)$, let $V_i(N') = \sum_{j \in N'} v_{i,j}$.

*Claim 3.9.* Fix a bidder $i$. For every $j \in \mathcal{D}(i)$, we have $\text{Spend}(j) + \text{Spend}(i) \geq B_i$.

Intuitively, if $\text{Spend}(j) < v_{i,j}$, then it must be that bidder $i$ could not bid higher because of their budget constraint, i.e., $B_i$ could be covered by $\text{Spend}(j) + \text{Spend}(i) \geq B_i$. See the full proof in the appendix.

*Claim 3.10.* Fix a bidder $i$. For every $j \in \mathcal{D}(i)$, we have $v_{i,j} \leq V_i$.

We prove this claim by contradiction. To that end assume there exists $j \in \mathcal{D}(i)$, such that $v_{i,j} > V_i$. For a sufficiently small $\varepsilon > 0$, if bidder $i$ changes their bid on query $j$ to $\text{Spend}(j) + \varepsilon$ and their bid on all other queries to 0, then bidder $i$ would win query $j$ and pay $\text{Spend}(j) + \varepsilon$. Their total value would be $v_{i,j} > V_i$. By the condition of Theorem 3.8, we know $v_{i,j} \leq B_i$. By the definition of $\mathcal{D}(i)$, $\text{Spend}(j) < v_{i,j}$ and thus, $\text{Spend}(j) + \varepsilon < v_{i,j}$ provided $\varepsilon$ is sufficiently small. Thus, bidder $i$ is able to get more value while satisfying both constraints, which contradicts the equilibrium assumption.

We further partition $A_{\bar{B}}$ into two sets:

$$A_{\bar{B}0} = \{i \in A_{\bar{B}} \mid \sum_{j \in \mathcal{D}(i)} \pi^*_{i,j} \geq 1\} \text{ and } A_{\bar{B}1} = A_{\bar{B}} \setminus A_{\bar{B}0}.$$

We bound $\text{OPT}(A_{\bar{B}0})$ and $\text{OPT}(A_{\bar{B}1})$ separately by Lemma 3.11 and Lemma 3.14.

**LEMMA 3.11.** *We have $\sum_{i \in A_{\bar{B}0}} \sum_{j \in \mathcal{D}(i)} \pi^*_{i,j} \cdot \text{Spend}(j) + \text{LW}(A_{\bar{B}0}) \geq \text{OPT}(A_{\bar{B}0})$.*

**PROOF.** Fix $i \in A_{\bar{B}0}$. By Claim 3.9, we have

$$\sum_{j \in \mathcal{D}(i)} \pi^*_{i,j} \cdot \text{Spend}(j) + \text{LW}(i)$$

$$\geq \sum_{j \in \mathcal{D}(i)} \pi^*_{i,j} \cdot \text{Spend}(j) + \text{Spend}(i) \geq B_i.$$

Summing up this inequality for all $i \in A_{\bar{B}0}$ yields

$$\sum_{i \in A_{\bar{B}0}} \sum_{j \in \mathcal{D}(i)} \pi^*_{i,j} \cdot \text{Spend}(j) + \text{LW}(A_{\bar{B}0}) \geq \sum_{i \in A_{\bar{B}0}} B_i \geq \text{OPT}(A_{\bar{B}0}).$$
□

To bound $\text{OPT}(A_{\bar{B}1})$, we require the following auxiliary lemma.

**LEMMA 3.12.** *Fix bidder $i \in A_{\bar{B}1}$. If $\sum_{j \in \mathcal{D}(i)} \pi^*_{i,j} < 1$ then*

$$\sum_{j \in N(i)} (v_{i,j} \cdot (1 - \pi^*_{i,j}) + \pi^*_{i,j} \cdot \text{Spend}(j))$$

$$\geq \sum_{j \in \mathcal{D}(i)} \pi^*_{i,j} \cdot (v_{i,j} - \text{Spend}(j)).$$

**PROOF.** We split $N(i)$ into two sets:

$$N_1 = \{j \in N(i) \mid \text{Spend}(j) \geq v_{i,j}\} \text{ and } N_2 = N(i) \setminus N_1.$$

We first show the following claim to bound $V_i(N_1) + \text{Spend}(N_2)$.

*Claim 3.13.* $\sum_{j \in N(i)} (v_{i,j} \cdot (1 - \pi^*_{i,j}) + \pi^*_{i,j} \cdot \text{Spend}(j)) \geq V_i(N_1) + \text{Spend}(N_2)$.

Fix a query $j' \in \mathcal{D}(i)$. Since bidder $i$ is in an equilibrium, it means bidder $i$ cannot get more value, while satisfying their constraints, by giving up queries in $N_1$ and bidding high enough to win query $j'$. In other words, if bidder $i$ deviates by bidding 0 on all queries in $N_1$ and bidding $\text{Spend}(j')$ on query $j'$ to win query $j'$ then at least one of the following two statements is true: (1) the value that bidder $i$ loses from $N_1$ is at least the value $i$ gains by winning $j'$ or (2) bidder $i$ violates their constraints. We analyze each case below.

 

*Case 1:* $v_{i,j'} \leq V_i(N_1)$. In this case, bidder $i$ gives up all queries in $N_1$ and bids $\text{SPEND}(j')$ to win $j'$, but bidder $i$'s total value decreases after the switch. In other words $V_i(N_1) \geq v_{i,j'} \geq v_{i,j'} - \text{SPEND}(j)$.

*Case 2: bidder $i$ violates at least one of their constraints.* It is not possible to violate their ROS constraint because $\text{SPEND}(N_1) \geq V_i(N_1)$ and $\text{SPEND}(j') < v_{i,j'}$. Thus bidder $i$ must violate their budget constraint. Bidder $i$ does not spend anything on $N_1$ now because bidder $i$ bids 0 on all queries in $N_1$. The total spend of bidder $i$ after the strategy deviation is $\text{SPEND}(j') + \text{SPEND}(N_2)$, which must be greater than $B_i$. Combining with the condition that $v_{i,j'} \leq B_i$, we have

$$\text{SPEND}(N_2) > B_i - \text{SPEND}(j') \geq v_{i,j'} - \text{SPEND}(j').$$

Combining Case 1 and Case 2 and Claim 3.13, we have that for every $j' \in \mathcal{D}(i)$:

$$\sum_{j \in N(i)} (v_{i,j} \cdot (1 - \pi_{i,j}^*) + \pi_{i,j}^* \cdot \text{SPEND}(j)) \geq V_i(N_1) + \text{SPEND}(N_2)$$

$$\geq v_{i,j'} - \text{SPEND}(j').$$

Since $\sum_{j' \in \mathcal{D}(i)} \pi_{i,j'}^* < 1$ and $v_{i,j} > \text{SPEND}(j')$ for $j \in \mathcal{D}(i)$, we conclude that

$$\sum_{j \in N(i)} v_{i,j} \cdot (1 - \pi_{i,j}^*) + \pi_{i,j}^* \cdot \text{SPEND}(j) \geq \sum_{j \in \mathcal{D}(i)} \pi_{i,j}^* \cdot (v_{i,j} - \text{SPEND}(j)).$$

$\square$

**LEMMA 3.14.** *We have*
$$\text{OPT}(A_{\bar{B}1}) \leq \text{LW}(A_{\bar{B}1}) + \sum_{i \in A_{\bar{B}1}} \sum_{j \in Q} \text{SPEND}(j) \cdot \pi_{i,j}^*$$

**PROOF.** Fix $i \in A_{\bar{B}1}$. We first split $\text{OPT}(i)$ into three parts and then bound each term separately. Indeed, for all $i \in A_{\bar{B}1}$, we have

$$\text{OPT}(i) \leq \sum_{j \in Q} v_{i,j} \cdot \pi_{i,j}^*$$

$$= \sum_{j \in N(i)} v_{i,j} \cdot \pi_{i,j}^* + \sum_{j \in \mathcal{D}(i)} v_{i,j} \cdot \pi_{i,j}^* + \sum_{j \in Q \setminus N(i) \setminus \mathcal{D}(i)} v_{i,j} \cdot \pi_{i,j}^*$$

$$= \sum_{j \in N(i)} v_{i,j} \cdot \pi_{i,j}^* + (\sum_{j \in \mathcal{D}(i)} (v_{i,j} - \text{SPEND}(j)) \cdot \pi_{i,j}^*$$

$$+ \sum_{j \in \mathcal{D}(i)} \text{SPEND}(j) \cdot \pi_{i,j}^*) + \sum_{j \in Q \setminus N(i) \setminus \mathcal{D}(i)} v_{i,j} \cdot \pi_{i,j}^*.$$

We know that if $j \in Q \setminus N(i) \setminus \mathcal{D}(i)$ we have $\text{SPEND}(j) \geq v_{i,j}$. Combining with Lemma 3.12 gives that

$$\text{OPT}(i)$$

$$\leq \sum_{j \in N(i)} v_{i,j} \cdot \pi_{i,j}^* + \sum_{j \in N(i)} (v_{i,j} \cdot (1 - \pi_{i,j}^*) + \pi_{i,j}^* \cdot \text{SPEND}(j))$$

$$+ \sum_{j \in \mathcal{D}(i)} \text{SPEND}(j) \cdot \pi_{i,j}^* + \sum_{j \in Q \setminus N(i) \setminus \mathcal{D}(i)} \text{SPEND}(j) \cdot \pi_{i,j}^*$$

$$= \sum_{j \in N(i)} v_{i,j} + \sum_{j \in Q} \text{SPEND}(j) \cdot \pi_{i,j}^*.$$

Summing up for all $i \in A_{\bar{B}1}$ gives that $\text{OPT}(A_{\bar{B}1}) \leq \text{LW}(A_{\bar{B}1}) + \sum_{i \in A_{\bar{B}1}} \sum_{j \in Q} \text{SPEND}(j) \cdot \pi_{i,j}^*.$ $\square$

We are ready to finish the proof by putting Inequality 5, Lemma 3.11 and Lemma 3.14 together. We have

$$\text{OPT} = \text{OPT}(A_B) + \text{OPT}(A_{\bar{B}0}) + \text{OPT}(A_{\bar{B}1})$$

$$\leq \text{LW}(A_B) + \text{LW}(A_{\bar{B}0}) + \sum_{i \in A_{\bar{B}0}} \sum_{j \in \mathcal{D}(i)} \pi_{i,j}^* \cdot \text{SPEND}(j)$$

$$+ \text{LW}(A_{\bar{B}1}) + \sum_{i \in A_{\bar{B}1}} \sum_{j \in Q} \text{SPEND}(j) \cdot \pi_{i,j}^*$$

$$\leq \text{LW}(A) + \sum_{i \in A} \sum_{j \in Q} \pi_{i,j}^* \cdot \text{SPEND}(j)$$

$$\leq \text{LW}(A) + \sum_{j \in Q} \sum_{i \in A} \pi_{i,j}^* \cdot \text{SPEND}(j)$$

$$\leq \text{LW}(A) + \sum_{j \in Q} \text{SPEND}(j) \leq 2\text{LW}(A),$$

which concludes that the PoA is at most 2. $\square$

## 3.1 Uniform bidding

It is known that, without budget constraints, when the bidders are assumed to bid uniformly, the PoA is 1 for FPA [16, Theorem 6.5]. In this section, we study uniform bidding when the bidders have a budget constraint. We first show, somewhat surprisingly, that the I-PoA of FPA now becomes $n$.

**THEOREM 3.15.** *If the bidders are assumed to bid uniformly in FPA with both ROS and budget constraints, then the integral price of anarchy is at least $n$.*

The following theorem gives a matching upper bound of the PoA of FPA with budget constraints.

**THEOREM 3.16.** *If the bidders are assumed to bid uniformly in FPA with both ROS and budget constraints, then the price of anarchy is at most $n$.*

## 4 RANDOMIZED FIRST-PRICE AUCTION

In this section, we study the efficiency of the randomized first-price auction [23, §5]. The auction is defined as follows. Given a parameter $\alpha \geq 1$ and the two highest bids $b_1 \geq b_2$:

- If $b_1 \geq \alpha b_2$ then bidder 1 wins with probability 1.
- Otherwise, bidder 1 wins with probability $\frac{1}{2} \left(1 + \log_\alpha(b_1/b_2)\right)$. With the remaining probability, bidder 2 wins.

The winner of the auction pays their bid. We use rFPA($\alpha$) to denote the randomized first-price auction with parameter $\alpha$. In this section, we focus on the setting where there are two bidders in each query.

We prove two results for rFPA when advertisers have a budget constraint. The first is that rFPA with an appropriate $\alpha$ parameter achieves at least 0.555 of the optimal liquid welfare; this is identical to the efficiency of rFPA when advertisers only have a ROS constraint [23].

**THEOREM 4.1.** *For any set of undominated bids for two bidders with both ROS and budget constraints, rFPA($\alpha = 1.4$) obtains at least $\frac{1}{1.8}$-fraction of the welfare of the optimal (randomized) allocation.*

For the second result, we consider the setting where advertisers are assumed to bid uniformly. Recall that Theorem 3.15 shows that uniform bidding increases the I-PoA for FPA. Interestingly, restricting bidders to bid uniformly actually lowers the PoA for rFPA. Intuitively, the reason that I-PoA is large for uniform bidding in FPA is because a bidder's bid to all queries are highly correlated. It is possible to construct instances where, using uniform bidding, a bidder can either win no queries or win everything and violate their budget constraint but it is not possible for the bidder to smoothly interpolate these extremes. However, in randomized auctions, bidders are able to smoothly increase or decrease their bids to win fractional queries, making it possible to avoid the bad cases in deterministic auctions.

**Theorem 4.2.** *If the bidders are assumed to bid uniformly in rFPA($\alpha = 7.62$), then the PoA is at most 1.5.*

## 5 QUASI-PROPORTIONAL FIRST PRICE AUCTION

In this section, we consider the following quasi-proportional power mechanism. Given a parameter $\alpha \geq 1$ and the bids $b_1, \ldots, b_n$, we allocate to bidder $i$ with probability $\frac{b_i^\alpha}{\|b\|_\alpha^\alpha}$, where $\|b\|_\alpha = (\sum_{i \in A} b_i^\alpha)^{1/\alpha}$. The winner of the auction is charged their bid. We note that a similar mechanism was also studied in [27] although they considered the mechanism with $\alpha \leq 1$.

**Theorem 5.1.** *The price of anarchy for the quasi-proportional FPA mechanism with both ROS and budget constraints is at most 2 when $\alpha \to \infty$.*

**Proof.** First, we require the following lemma.

**Lemma 5.2.** *Fix a query $j$ and suppose that $0 < \Pr[i \text{ wins query } j] \leq \eta$. Then*
$$\text{SPEND}(j) \geq v_{i,j} \cdot \frac{\alpha \cdot (1 - \eta)}{(n\eta)^{1/\alpha}(\alpha - \alpha\eta + 1)}.$$

**Lemma 5.3.** *For every $i \in A_{\bar{B}}$, $j \in Q$, $\eta \in (0, 1)$, and $y_{i,j} \in [0, 1]$, we have*
$$\pi_{i,j} \cdot v_{i,j} + y_{i,j} \cdot \frac{(n\eta)^{1/\alpha}\eta(\alpha - \alpha\eta + 1)}{\alpha(1 - \eta)} \cdot \text{SPEND}(j) \geq \eta \cdot y_{i,j}v_{i,j}.$$

**Proof.** If $\pi_{i,j} \geq \eta$ then $\pi_{i,j} \cdot v_{i,j} \geq \eta \cdot y_{i,j}v_{i,j}$. Otherwise, if $\pi_{i,j} < \eta$ then we can apply Lemma 5.2. □

We now apply Lemma 5.3 with $y_{i,j} = \frac{\pi_{i,j}^*}{\sum_{i' \in A_{\bar{B}}} \pi_{i',j}^*}$. For a fixed bidder $i$, we can sum over all $j$ to get that

$$\eta \cdot \text{OPT}(i) = \eta \sum_{j \in Q} \pi_{i,j}^* v_{i,j} \leq \sum_{j \in Q} \eta \cdot \frac{\pi_{i,j}^*}{\sum_{i' \in A_{\bar{B}}} \pi_{i',j}^*} v_{i,j}$$

$$\leq \sum_{j \in Q} \frac{\pi_{i,j}^*}{\sum_{i' \in A_{\bar{B}}} \pi_{i',j}^*} \frac{(n\eta)^{1/\alpha}\eta(\alpha - \alpha\eta + 1)}{\alpha(1 - \eta)} \cdot \text{SPEND}(j)$$

$$+ \sum_{j \in Q} \pi_{i,j}v_{i,j}$$

$$\leq \frac{(n\eta)^{1/\alpha}\eta(\alpha - \alpha\eta + 1)}{\alpha(1 - \eta)} \cdot \sum_{j \in Q} \frac{\pi_{i,j}^*}{\sum_{i' \in A_{\bar{B}}} \pi_{i',j}^*} \text{SPEND}(j)$$

$+ \text{LW}(i)$.

Thus, summing both sides over $i \in A_{\bar{B}}$, we have that

$$\eta \cdot \text{OPT}(A_{\bar{B}})$$

$$\leq \text{LW}(A_{\bar{B}}) + \frac{(n\eta)^{1/\alpha}\eta(\alpha - \alpha\eta + 1)}{\alpha(1 - \eta)} \cdot \sum_{j \in Q} \text{SPEND}(j)$$

$$\leq \text{LW}(A_{\bar{B}}) + \frac{(n\eta)^{1/\alpha}\eta(\alpha - \alpha\eta + 1)}{\alpha(1 - \eta)} \cdot \text{LW}.$$

Adding $\text{LW}(A_B) = \text{OPT}(A_B)$ to both sides gives that $\eta \cdot \text{OPT} \leq \left(1 + \frac{(n\eta)^{1/\alpha}\eta(\alpha - \alpha\eta + 1)}{\alpha(1 - \eta)}\right) \cdot \text{LW}$. Thus, the PoA is at most $\frac{1}{\eta} + \frac{(n\eta)^{1/\alpha}(\alpha - \alpha\eta + 1)}{\alpha(1 - \eta)}$. If we take limits as $\alpha \to \infty$, we get that this converges to $1/\eta + 1$. Since $\eta$ is arbitrary, we conclude that the PoA is 2. □

## 6 DISCUSSION

In this paper, we study the PoA and I-POA of auto-bidders with both budget and ROS constraints in non-truthful auctions, which complement previous works on the PoA of different auctions for auto-bidders with only ROS constraints. The setting with budget constraints is different and challenging for several reasons. First, Theorem 3.1 shows that there is a large gap between deterministic and randomized allocations in our setting. In contrast, recall that the best deterministic allocation is also the best randomized allocation in the setting with only a ROS constraint. Thus, the techniques required for the efficiency analyses are different from previous related work and may be extended for future work with budget constraints. Second, when the bidders are assumed to bid uniformly, our results are surprisingly inconsistent with setting with only a ROS constraint. Deng et al. [16] shows that the PoA is 1 when bidders are assumed to bid uniformly in FPA with only ROS constraints. However, we show that the I-PoA for this setting with budget constraints becomes $n$, which is worse than the I-PoA of 2 for non-uniform bidding. We also show that uniform bidding does improve the PoA for randomized FPA in Theorem 4.2. Intuitively, the reason that I-PoA is large for uniform bidding in FPA is because a bidder's bid to all queries are highly correlated, and it is possible that a bidder could not win any query because when they will have to take none or too many queries and violate their budget constraint. However, in randomized auctions, bidders are able to smoothly increase or decrease their bids to win fractional queries to avoid such bad cases in deterministic auctions.

There are many interesting future directions to explore beyond this work. We analyze the PoA and I-PoA of different non-truthful auctions for the cases that a Nash equilibrium exists. One question we leave for future work the existence of Nash equilibria. For the budget constraints in randomized auctions, we study the ex ante version. It will be interesting to also consider the ex post budget constraint setting and compare it with our setting.

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

# A MISSING PROOFS FOR SECTION 3

## A.1 Proof of Theorem 3.6

PROOF OF THEOREM 3.6. We split the bidders into two sets:

$$A_B = \{i \in A \mid B_i \leq \sum_{j \in N(i)} v_{i,j}\} \text{ and } A_{\bar{B}} = A \setminus A_B.$$

Note that, by the definition of liquid welfare, for $i \in A_B$, we have $\text{LW}(i) = B_i \geq \text{I-OPT}(i)$. Thus, we have

$$\text{LW}(A_B) \geq \text{I-OPT}(A_B). \tag{6}$$

We further split $A_{\bar{B}}$ in to two sets. We define $A_{\bar{B}1}$ as the set of bidders $i$ for which there exists at least one query $j \in O(i) \setminus N(i)$ such that if bidder $i$ wins query $j$ as well then their total value would be at least $B_i$. In other words, if bidder $i$ gets this extra query $j$, then they will achieve their optimal liquid welfare $B_i$. Next, we define $A_{\bar{B}0} = A_{\bar{B}} \setminus A_{\bar{B}1}$. These are the bidders for which we can add any query in $O(i) \setminus N(i)$ and their total value would still be at most $B_i$. Formally, the sets are defined as

$$A_{\bar{B}1} = \{i \in A_{\bar{B}} \mid \exists j \in O(i) \setminus N(i), s.t. \ v_{i,j} + \sum_{j' \in N(i)} v_{i,j'} \geq B_i\}$$

$$A_{\bar{B}0} = A_{\bar{B}} \setminus A_{\bar{B}1}.$$

We first bound $\text{I-OPT}(A_{\bar{B}1})$ and $\text{I-OPT}(A_{\bar{B}0})$ in Lemma A.1 and Lemma A.2 below, and then put everything together using the constraint that the total spend in the auction is upper bounded by the total liquid welfare.

LEMMA A.1. *We have*
$\sum_{i \in A_{\bar{B}1}} \sum_{j \in O(i) \setminus N(i)} \text{SPEND}(j) + \text{LW}(A_{\bar{B}1}) \geq \text{I-OPT}(A_{\bar{B}1}).$

PROOF. Fix any bidder $i \in A_{\bar{B}1}$ and consider a query $j' \in O(i) \setminus N(i)$ such that $v_{i,j} + \sum_{j \in N(i)} v_{i,j} \geq B_i$ (such a query $j'$ must exist by definition of $A_{\bar{B}1}$). We claim that $\text{SPEND}(j') \geq B_i - \sum_{j \in N(i)} v_{i,j}$ in any equilibrium of FPA. We will prove this by contradiction, so assume $\text{SPEND}(j') < B_i - \sum_{j \in N(i)} v_{i,j}$. Then the highest bid on $j'$ must be at most $B_i - \sum_{j \in N(i)} v_{i,j}$. Now, observe that if bidder $i$ bids $B_i - \sum_{j \in N(i)} v_{i,j}$ on query $j'$ then bidder $i$ would win query $j'$. Their total value would then be $B_i$ and their total spend would be at least $B_i$. Thus their value has increased while both their budget and ROS constraints are satisfied. This contradicts the assumption that the bidders are in equilibrium. We conclude that $\text{SPEND}(j') \geq B_i - \sum_{j \in N(i)} v_{i,j}$ for all bidders $i \in A_{\bar{B}1}$. Using the trivial upper bound $\text{SPEND}(j') \leq \sum_{j \in O(i) \setminus N(i)} \text{SPEND}(j)$ (since $j' \in O(i) \setminus N(i)$), this implies that $\sum_{j \in O(i) \setminus N(i)} \text{SPEND}(j) + \sum_{j \in N(i)} v_{i,j}$ for $i \in A_{\bar{B}1}$. We thus have that,

$$\text{I-OPT}(A_{\bar{B}1}) \leq \sum_{i \in A_{\bar{B}1}} B_i$$

$$\leq \sum_{i \in A_{\bar{B}1}} \sum_{j \in O(i) \setminus N(i)} \text{SPEND}(j) + \sum_{i \in A_{\bar{B}1}} \sum_{j \in N(i)} v_{i,j}$$

$$\leq \sum_{i \in A_{\bar{B}1}} \sum_{j \in O(i) \setminus N(i)} \text{SPEND}(j) + \text{LW}(A_{\bar{B}1}),$$

as claimed. Note that the first line is because $\text{I-OPT}(A_{\bar{B}1}) = \sum_{i \in A_{\bar{B}1}} \min\{B_i, \sum_{j \in O(i)} v_{i,j}\} \leq \sum_{i \in A_{\bar{B}1}} B_i.$ □

LEMMA A.2. *We have*
$\sum_{i \in A_{\bar{B}0}} \sum_{j \in O(i) \setminus N(i)} \text{SPEND}(j) + \text{LW}(A_{\bar{B}0}) \geq \text{I-OPT}(A_{\bar{B}0}).$

PROOF. The proof is similar to the proof of Lemma A.1. For $i \in A_{\bar{B}0}$, we claim that for each query $j' \in O(i) \setminus N(i)$, $\text{SPEND}(j')$ is at least $v_{i,j'}$. Again, we prove this by contradiction. Assume there exists $j' \in O(i) \setminus N(i)$, such that $\text{SPEND}(j') < v_{i,j'}$. Then bidder $i$ can bid $v_{i,j'}$ to win query $j'$. Note that

$$c_{i,j'} + \sum_{j \in N(i)} c_{i,j} \leq v_{i,j'} + \sum_{j \in N(i)} c_{i,j} < B_i,$$

where the first inequality uses that bidder $i$ bids and pays $v_{i,j'}$ on query $j'$ and that bidder $i$'s ROS constraint was initially satisfied and the second is by definition of $A_{\bar{B}0}$. But this shows that bidder $i$ can improve their value without violating their constraints, which contradicts the assumption that the bidders are in an equilibrium. Thus, it must be $\text{SPEND}(j') \geq v_{i,j'}$ for all $i \in A_{\bar{B}0}$ and $j' \in O(i) \setminus N(i)$. In particular, $\sum_{j \in O(i) \setminus N(i)} \text{SPEND}(j) \geq \sum_{j \in O(i) \setminus N(i)} v_{i,j}$. Following a similar argument as in Lemma A.1, we have that

$$\text{I-OPT}(A_{\bar{B}0}) \leq \sum_{i \in A_{\bar{B}0}} \sum_{j \in O(i)} v_{i,j}$$

$$= \sum_{i \in A_{\bar{B}0}} \sum_{j \in O(i) \setminus N(i)} v_{i,j} + \sum_{i \in A_{\bar{B}0}} \sum_{j \in N(i)} v_{i,j}$$

$$\leq \sum_{i \in A_{\bar{B}0}} \sum_{j \in O(i) \setminus N(i)} \text{SPEND}(j) + \text{LW}(A_{\bar{B}0}).$$

The first inequality is because $\text{I-OPT}(A_{\bar{B}0}) = \sum_{i \in A_{\bar{B}0}} \min\{B_i, \sum_{j \in O(i)} v_{i,j}\} \leq \sum_{i \in A_{\bar{B}0}} \sum_{j \in O(i)} v_{i,j}.$ □

We now return to the proof of Theorem 3.6. Since each bidder's spend is at most their liquid welfare, we have that

$$\text{LW}(A) \geq \sum_{i \in A} \sum_{j \in O(i)} \text{SPEND}(j)$$

$$\geq \sum_{i \in A_{\bar{B}1}} \sum_{j \in O(i) \setminus N(i)} \text{SPEND}(j)$$

$$+ \sum_{i \in A_{\bar{B}0}} \sum_{j \in O(i) \setminus N(i)} \text{SPEND}(j)$$

Combining Inequality (6), Lemma A.1 and Lemma A.2, we have

$$\text{I-OPT} = \text{I-OPT}(A_B) + \text{I-OPT}(A_{\bar{B}1}) + \text{I-OPT}(A_{\bar{B}0})$$

$$\leq \text{LW}(A_B) + \sum_{i \in A_{\bar{B}1}} \sum_{j \in O(i) \setminus N(i)} \text{SPEND}(j) + \text{LW}(A_{\bar{B}1})$$

$$+ \text{LW}(A_{\bar{B}0}) + \sum_{i \in A_{\bar{B}0}} \sum_{j \in O(i) \setminus N(i)} \text{SPEND}(j)$$

$$\leq \text{LW}(A_B) + \text{LW}(A_{\bar{B}1}) + \text{LW}(A_{\bar{B}0}) + \text{LW}(A)$$

$$= 2\text{LW}(A),$$

which completes the proof. □

## A.2 Proof of Claim 3.9

PROOF OF CLAIM 3.9. We prove this claim by contradiction. To that end, assume there exists $j \in \mathcal{D}(i)$ such that $\text{SPEND}(j) + \text{SPEND}(i) < B_i$. Since we are using a FPA, this means the highest bid on query $j$ is exactly $\text{SPEND}(j)$. Recall that $\text{SPEND}(j) <$

$v_{i,j}$ by the definition of $\mathcal{D}(i)$. Let $\varepsilon$ be such that $0 < \varepsilon < \min\{B_i - \text{Spend}(j) - \text{Spend}(i), v_{i,j} - \text{Spend}(j)\}$. Thus bidder $i$ can win query $j$ by bidding and spending $\text{Spend}(j) + \varepsilon$ without violating their budget constraint. Note that bidder $i$ also satisfies their ROS constraint after winning $j$. We conclude that bidder $i$ is able to obtain more value while satisfying their constraints, which contradicts the equilibrium assumption. □

## A.3 Proof of Claim 3.10

PROOF OF CLAIM 3.10. We prove this claim by contradiction. To that end assume there exists $j \in \mathcal{D}(i)$, such that $v_{i,j} > V_i$. For a sufficiently small $\varepsilon > 0$, if bidder $i$ changes their bid on query $j$ to $\text{Spend}(j) + \varepsilon$ and their bid on all other queries to 0, then bidder $i$ would win query $j$ and pay $\text{Spend}(j) + \varepsilon$. Their total value would be $v_{i,j} > V_i$. By the condition of Theorem 3.8, we know $v_{i,j} \leq B_i$. By the definition of $\mathcal{D}(i)$, $\text{Spend}(j) < v_{i,j}$ and thus, $\text{Spend}(j) + \varepsilon < v_{i,j}$ provided $\varepsilon$ is sufficiently small. Thus, bidder $i$ is able to get more value while satisfying both constraints, which contradicts the equilibrium assumption. □

## A.4 Proof of Claim 3.13

PROOF OF CLAIM 3.13. Observe that

$$v_{i,j} \cdot (1 - \pi_{i,j}^*) + \pi_{i,j}^* \cdot \text{Spend}(j) \geq \begin{cases} v_{i,j} & j \in N_1 \\ \text{Spend}(j) & j \in N_2 \end{cases}.$$

The claim follows by summing this inequality for $j \in N(i)$. □

## A.5 Proof of Theorem 3.15

PROOF OF THEOREM 3.15. We construct the instance as follows. There are $n$ bidders $\{1, \ldots, n\}$ and $n$ queries $\{1, \ldots, n\}$. Bidder 1 has $B_1 = +\infty$, $v_{1,1} = b_{1,1} = 1 + \varepsilon$, and for every $j \in \{2, \ldots, n\}$, $v_{1,j} = b_{1,j} = 2\varepsilon$. Note that the uniform multiplier of bidder 1 is 1. Next, for every $i \in \{2, \ldots, n\}$, suppose bidder $i$ has $B_i = 1$, $v_{i,1} = \frac{1}{\varepsilon}$, and $v_{i,i} = 1$. All other values are 0. We will discuss the bids of bidder $2, \ldots, n$ below.

Note if bidder 1 wins every query then they have no incentive to change their bid since they are paying their value on each query. We now show that, if bidder 1 does use a uniform multiplier of 1 then in any equilibrium, bidder $i \in \{2, \ldots, n\}$ cannot win any query. Indeed, note that they cannot win query 1 because they would need to beat bidder 1 and pay at least $1 + \varepsilon$, violating their budget constraint. Since their value for query 1 is $1/\varepsilon$, this means bidder $i$ must use a bid multiplier $m_i \leq \varepsilon(1 + \varepsilon)$. Next, recall that their value for query $i$ is 1. And thus, they bid at most $\varepsilon(1 + \varepsilon) < 2\varepsilon$ since $\varepsilon < 1$ on query $i$. Thus, bidder $i$ loses this query as well.

Based on the above analysis, the only feasible allocation in any equilibrium is that bidder 1 wins all the queries, which has a total liquid welfare of $1 + \varepsilon + (n - 1) \cdot 2\varepsilon$. The optimal (integral) allocation would be to allocate query $i$ to bidder $i$ for $i \in \{1, \ldots, n\}$. The optimal (integral) liquid welfare is $\text{I-Opt} = 1 + \varepsilon + (n - 1) = n + \varepsilon$. Thus, the liquid welfare at equilibrium is at most $\frac{1 + \varepsilon + 2(n-1)\varepsilon}{n + \varepsilon} \cdot \text{I-Opt} \leq \left( \frac{1}{n} + 3\varepsilon \right) \cdot \text{I-Opt}$. Replacing $\varepsilon$ with $\varepsilon/3$ in the argument proves the theorem. □

## A.6 Proof of Theorem 3.16

PROOF OF THEOREM 3.16. Suppose that each bidder $i$ has a uniform bid multiplier $m_i$, i.e. for every $j \in Q$, $b_{i,j} = m_i \cdot v_{i,j}$. Given the bidders' multipliers $m_1, m_2, \ldots, m_n$, let $\pi_i(m_1, m_2, \ldots, m_n) \in [0, 1]^Q$ denote bidder $i$'s allocation and $c_i(m_1, m_2, \ldots, m_n) \in \mathbb{R}$ denote bidder $i$'s total spend across all queries. In an equilibrium, the spend must be feasible, i.e., $c_i(m_1, m_2, \ldots, m_n) \leq \min\{B_i, \sum_{j \in N(i)} v_{i,j}\}$.

Consider an equilibrium EQ. Similar to the proof of Theorem 3.6, we split the bidders into two sets:

$$A_B = \{i \in A \mid B_i \leq \sum_{j \in N(i)} v_{i,j}\} \text{ and } A_{\bar{B}} = A \setminus A_B.$$

Observe that for $i \in A_B$, we have $\text{LW}(i) = B_i \geq \text{Opt}(i)$. Summing up for all $i \in A_B$, we have

$$\text{LW}(A_B) \geq \text{Opt}(A_B). \tag{7}$$

We further partition $A_{\bar{B}}$ into two sets:

$$A_{\bar{B}0} = \{i \in A_{\bar{B}} \mid \pi_i(m_i, m_{-i}) = \pi_i(1, m_{-i}) \text{ and } c_i(1, m_{-i}) \leq B_i\}$$

$$A_{\bar{B}1} = A_{\bar{B}} \setminus A_{\bar{B}0}.$$

In words, $A_{\bar{B}0}$ is the set of bidders $i$ such that either (i) $m_i = 1$ or (ii) if bidder $i$ changes their bid multiplier $m_i$ to 1 then they win the same set of queries as in EQ and still satisfy their budget constraint. We will bound $\text{Opt}(A_{\bar{B}1})$ and $\text{Opt}(A_{\bar{B}0})$ in Lemma A.4 and Lemma A.6 below, and then put everything together using the constraint that the total spend in the auction is upper bounded by the total liquid welfare.

We first show the following claim when $m_i > 1$ for any bidder $i$.

*Claim A.3.* If the bidders are assumed to bid uniformly in FPA and $m_i > 1$ in an equilibrium EQ then bidder $i$ does not win any query. In particular, $i \in A_{\bar{B}0}$. In addition, $\text{Spend}(j) \geq v_{i,j}$ for every $j \in Q$.

PROOF. The first assertion follows from the observation that if bidder $i$ did win any queries then their total cost would be more than their total value. Thus, bidder $i$ would violate their ROS constraint. Since the allocation is monotone in a bidder's bid, bidder $i$ would still win no query when $m_i$ is lowered to 1, i.e., $\pi_i(m_i, m_{-i}) = \pi_i(m_i' = 1, m_{-i})$. Hence, $i \in A_{\bar{B}0}$.

For the last assertion, note that on each query $j$, the winner in EQ bids and pays at least $m_i \cdot v_{i,j} > v_{i,j}$. □

LEMMA A.4. *We have*
$$\text{LW}(A_{\bar{B}0}) + \sum_{i \in A_{\bar{B}0}} \sum_{j \notin N(i)} \text{Spend}(j) \geq \text{Opt}(A_{\bar{B}0}).$$

PROOF. We first show that for all $i \in A_{\bar{B}0}$ and $j \notin N(i)$, we have $\text{Spend}(j) \geq v_{i,j}$ and thus,

$$\sum_{j \notin N(i)} \text{Spend}(j) \geq \sum_{j \notin N(i)} v_{i,j}. \tag{8}$$

To see this, fix $i \in A_{\bar{B}0}$. If $m_i > 1$ then Claim A.3 shows that $\text{Spend}(j) \geq v_{i,j}$. Now suppose $m_i \leq 1$. By definition of $A_{\bar{B}0}$, for every query $j \notin N(i)$, if bidder $i$ increases their bid to $v_{i,j}$, they would continue to lose that query. This implies that the winner of $j \notin N(i)$ pays at least $v_{i,j}$.

Since $i \in A_{\bar{B}0}$, we have that $\mathrm{LW}(i) = \sum_{j \in N(i)} v_{i,j}$. Combining with Eq. (8), we get that

$$\mathrm{LW}(i) + \sum_{j \notin N(i)} \mathrm{SPEND}(j) \geq \sum_{j \in N(i)} v_{i,j} + \sum_{j \notin N(i)} v_{i,j}$$
$$= \sum_{j \in Q} v_{i,j} \geq \mathrm{OPT}(i).$$

The last inequality holds because $\sum_{j \in Q} v_{i,j}$ is an upper bound on bidder $i$'s liquid welfare. Summing up the above inequality over all $i \in A_{\bar{B}0}$, we conclude that

$$\mathrm{LW}(A_{\bar{B}0}) + \sum_{i \in A_{\bar{B}0}} \sum_{j \notin N(i)} \mathrm{SPEND}(j) \geq \mathrm{OPT}(A_{\bar{B}0}). \qquad \square$$

LEMMA A.5. *For every bidder $i \in A_{\bar{B}1}$, we have that $\pi_i(1, m_{-i}) \neq \pi_i(m_i, m_{-i})$ and $c_i(1, m_{-i}) > B_i$.*

PROOF. If $i \in A_{\bar{B}1}$ then there can be three possibilities. Either (1) $\pi_i(1, m_{-i}) = \pi_i(m_i, m_{-i})$ and $c_i(1, m_{-i}) > B_i$, (2) $\pi_i(1, m_{-i}) \neq \pi_i(m_i, m_{-i})$ and $c_i(1, m_{-i}) \leq B_i$, or (3) $\pi_i(1, m_{-i}) \neq \pi_i(m_i, m_{-i})$ and $c_i(1, m_{-i}) > B_i$. We show that the first two are impossible and so the last condition must hold.

To see that (1) is not possible, note that since $i \notin A_B$ we have $B_i > \sum_{j \in N(i)} v_{i,j} = \sum_{j \in Q} \pi_{i,j}(m_i, m_{-i}) v_j$. Thus, if $\pi_i(1, m_{-i}) = \pi_i(m_i, m_{-i})$ then we would have $B_i > \sum_{j \in Q} \pi_{i,j}(1, m_{-i}) v_j = c_i(1, m_{-i})$.

To see that (2) is not possible, note that Claim A.3 shows that $m_i \leq 1$ and the fact that $i \in A_{\bar{B}1}$ means that $m_i \neq 1$. Thus, $m_i < 1$. Therefore, if $\pi_i(1, m_{-i}) \neq \pi_i(m_i, m_{-i})$ and $c_i(1, m_{-i}) \leq B_i$ then the multipliers would not form an equilibrium (since the allocation is non-decreasing in bid). $\qquad \square$

For $i \in A_{\bar{B}1}$, define $m'_i = \inf\{m \in (m_i, 1] \mid \pi_i(m, m_{-i}) \neq \pi_i(m_i, m_{-i})$ and $c_i(m, m_{-i}) > B_i\}$. Note that $m'_i$ is well-defined (and in $(m_i, 1]$) by Lemma A.5.

LEMMA A.6. *Let $\varepsilon > 0$ be such that $m_i + \varepsilon/2 < m'_i$ for every $i \in A_{\bar{B}1}$. Then*

$$\mathrm{LW}(A_{\bar{B}1}) + \sum_{i \in A_{\bar{B}1}} \sum_{j \notin N(i)} \mathrm{SPEND}(j) + \sum_{i \in A_{\bar{B}1}} \sum_{j \notin N(i)} \varepsilon \cdot v_{i,j}$$
$$\geq \mathrm{OPT}(A_{\bar{B}1}).$$

PROOF. Consider a bidder $i \in A_{\bar{B}1}$. Let $m''_i = \min\{1, m'_i + \varepsilon/2\}$. By the definition of $m'_i$ and Lemma A.5, if bidder $i$ uses a bid multiplier of $m''_i$ then bidder $i$ wins strictly more queries but must then violate their budget constraint. We thus have that

$$B_i < \sum_{j \in N'(i)} m''_i \cdot v_{i,j} \qquad (9)$$
$$= \sum_{j \in N(i)} m''_i \cdot v_{i,j} + \sum_{j \in N'(i) \setminus N(i)} m''_i \cdot v_{i,j}$$
$$\leq \sum_{j \in N(i)} v_{i,j} + \sum_{j \in N'(i) \setminus N(i)} (m'_i + \varepsilon/2) \cdot v_{i,j} \qquad (10)$$
$$= \mathrm{LW}(i) + \sum_{j \in N'(i) \setminus N(i)} (m'_i + \varepsilon/2) \cdot v_{i,j}. \qquad (11)$$

Since $m_i + \varepsilon/2 < m'_i$, it must be that bidder $i$ still wins $N(i)$ when setting their multiplier to $m'_i - \varepsilon$. In other words, the spend of query $j \in N'(i) \setminus N(i)$ must be at least $(m'_i - $

$\varepsilon/2) v_{i,j}$. Combining with Eq. (11), we have $B_i < \mathrm{LW}(i) + \sum_{j \in N'(i) \setminus N(i)} (\mathrm{SPEND}(j) + \varepsilon \cdot v_{i,j})$. Therefore, using that $\mathrm{OPT}(i) \leq B_i$, we have $\mathrm{OPT}(i) \leq \mathrm{LW}(i) + \sum_{j \notin N(i)} \mathrm{SPEND}(j) + \sum_{j \notin N(i)} \varepsilon \cdot v_{i,j}$. Summing this inequality for all $i \in A_{\bar{B}1}$, we get that

$$\mathrm{OPT}(A_{\bar{B}1}) \leq \mathrm{LW}(A_{\bar{B}1}) + \sum_{i \in A_{\bar{B}1}} \sum_{j \notin N(i)} \mathrm{SPEND}(j)$$
$$+ \sum_{i \in A_{\bar{B}1}} \sum_{j \notin N(i)} \varepsilon \cdot v_{i,j}. \qquad \square$$

Let $N^{-1}(j)$ denote the advertiser that wins query $j$ and let $V = \max_{i \in A} \sum_{j \in Q} v_{i,j}$. Combining Lemma A.6 and Lemma A.4, we get that

$$\mathrm{OPT}(A_{\bar{B}}) = \mathrm{OPT}(A_{\bar{B}0}) + \mathrm{OPT}(A_{\bar{B}1})$$
$$\leq \mathrm{LW}(A_{\bar{B}0}) + \sum_{i \in A_{\bar{B}0}} \sum_{j \notin N(i)} \mathrm{SPEND}(j) + \mathrm{LW}(A_{\bar{B}1})$$
$$+ \sum_{i \in A_{\bar{B}1}} \sum_{j \notin N(i)} \mathrm{SPEND}(j) + \sum_{i \in A_{\bar{B}1}} \sum_{j \notin N(i)} \varepsilon \cdot v_{i,j}$$
$$\leq \mathrm{LW}(A_{\bar{B}}) + \sum_{i \in A_{\bar{B}}} \sum_{j \notin N(i)} \mathrm{SPEND}(j) + \varepsilon n V$$
$$\leq \mathrm{LW}(A_{\bar{B}}) + \sum_{i \in A} \sum_{j \notin N(i)} \mathrm{SPEND}(j) + \varepsilon n V$$
$$\leq \mathrm{LW}(A_{\bar{B}}) + \sum_{j \in Q} \sum_{i \in A \setminus N^{-1}(j)} \mathrm{SPEND}(j) + \varepsilon n V \quad (12)$$
$$\leq \mathrm{LW}(A_{\bar{B}}) + (n-1) \cdot \sum_{j \in Q} \mathrm{SPEND}(j) + \varepsilon n V \quad (13)$$
$$\leq \mathrm{LW}(A_{\bar{B}}) + (n-1) \cdot \mathrm{LW}(A) + \varepsilon n V. \quad (14)$$

In Eq. (12), we swapped the order of the sum. In Eq. (13), we used that $|A \setminus N^{-1}(j)| = n - 1$ (each query is assigned to exactly one bidder). In Eq. (14), we used that the liquid welfare is an upper bound on the total spend. Finally, combining with Eq. (7), we conclude that

$$\mathrm{OPT}(A) = \mathrm{OPT}(A_B) + \mathrm{OPT}(A_{\bar{B}})$$
$$\leq \mathrm{LW}(A_B) + \mathrm{LW}(A_{\bar{B}}) + (n-1) \cdot \mathrm{LW}(A) + \varepsilon n V$$
$$= n \cdot \mathrm{LW}(A) + \varepsilon n V.$$

Since the above inequality is true for every $\varepsilon > 0$, we conclude that $\mathrm{OPT}(A) \leq n \cdot \mathrm{LW}(A)$. $\qquad \square$

# B MISSING PROOFS FOR SECTION4

## B.1 Proof of Theorem 4.1

PROOF OF THEOREM 4.1. Let $\pi_{i,j}$ denote the probability that bidder $i$ wins query $j$ in an equilibrium, $\pi^*_{i,j}$ be the probability that bidder $i$ wins query $j$ for some $\pi^* \in \mathrm{argmax}_{\pi \in \Pi}\{\mathrm{LW}(\pi)\}$. Let $\tilde{v}_{i,j} = \pi_{i,j} \cdot v_{i,j}$ be the expected value bidder $i$ obtains from query $j$ in the equilibrium and $v^*_{i,j} = \pi^*_{i,j} \cdot v_{i,j}$ be the expected value that bidder $i$ obtains from query $j$ in OPT. For a fixed equilibrium EQ, let $V_i^{\mathrm{EQ}} = \sum_{j \in Q} \pi_{i,j} v_{i,j}$ be the value that bidder $i$ obtains in equilibrium. For a bidder $i$, let $\bar{i}$ denote the other bidder.

**Lemma B.1.** *Suppose that bidder $i$ wins query $j$ with probability 1. If $V_{\bar{i}}^{\mathrm{EQ}} < B_{\bar{i}}$ and the bids are undominated then $b_{i,j} \geq \alpha v_{\bar{i},j}$.*

**Proof.** We will prove the contrapositive statement. In other words, we will show that if $V_{\bar{i}}^{\mathrm{EQ}} < B_{\bar{i}}$ and $b_{i,j} < \alpha v_{\bar{i},j}$ then the bids are not undominated. To that end, suppose that bidder $i$ wins query $j$ with probability 1 and $b_{i,j} < \alpha v_{\bar{i},j}$. Since bidder $i$ wins with probability 1, it must be that $b_{\bar{i},j} \leq b_{i,j}/\alpha$. Let $\mathrm{Cost}(b) = \frac{b}{2}(1 + \log_\alpha \frac{b}{b_{i,j}})$ be the cost function of bidder $\bar{i}$'s bid $b = b_{\bar{i},j}$. We consider two cases based on whether $\mathrm{Cost}(v_{\bar{i},j}) \leq B - V_{\bar{i}}^{\mathrm{EQ}}$ or $\mathrm{Cost}(v_{\bar{i},j}) > B - V_{\bar{i}}^{\mathrm{EQ}}$.

*Case 1:* $\mathrm{Cost}(v_{\bar{i},j}) \leq B - V_{\bar{i}}^{\mathrm{EQ}}$. Bidder $\bar{i}$ could bid $v_{\bar{i},j}$ and have a non-zero probability of winning while maintaining both their ROS and budget constraints. Indeed, since $b_{\bar{i},j} = v_{\bar{i},j} > b_{i,j}/\alpha$, this would mean that bidder $\bar{i}$ wins with some probability $p \in (0,1]$. In this case, bidder $\bar{i}$'s value would increase by $p \cdot v_{\bar{i},j} > 0$. Moreover, they pay $\mathrm{Cost}(v_{\bar{i},j}) = p \cdot v_{\bar{i},j}$ which is less than both the value gained and their remaining budget.

*Case 2:* $\mathrm{Cost}(v_{\bar{i},j}) > B - V_{\bar{i}}^{\mathrm{EQ}}$. Observe that $\mathrm{Cost}(b)$ is a non-decreasing continuous function of $b$ with $\mathrm{Cost}(b_{i,j}/\alpha) = 0$ and $\mathrm{Cost}(v_{\bar{i},j}) > 0$. Hence, there exists $\hat{b} \in (b_{i,j}/\alpha, v_{\bar{i},j})$ such that $0 < \mathrm{Cost}(\hat{b}) \leq B - V_{\bar{i}}^{\mathrm{EQ}}$. Similar to Case 1, if bidder $\bar{i}$ bids $\hat{b}$ then bidder $\bar{i}$ wins with some probability $p \in (0,1]$. Their value would increase by $p \cdot v_{\bar{i},j} > 0$ and they would pay $\mathrm{Cost}(\hat{b}) = p \cdot \hat{b} < p \cdot v_{\bar{i},j}$. In particular, bidder $\bar{i}$ would continue to satisfy both their ROS and budget constraints.

To conclude, in both Case 1 and Case 2, bidder $\bar{i}$ can increase their value while maintaining both their ROS and budget constraints, which contradicts the fact that the bids are undominated. □

We restate the following lemma in [23]:

**Lemma B.2** ([23, Lemma 5.5]). *Fix bidder $i$ and query $j$. For any set of undominated bids, if $\pi_{i,j} \in (0,1)$ and $V_i < B_i$ then $b_{i,j} \geq \frac{v_{i,j}}{1+\ln(\alpha)+\ln(\beta_i)}$ where $\beta_{i,j} = b_{i,j}/b_{\bar{i},j}$ is the ratio between bidder $i$'s bid and the other bidder's bid.*

We follow the proof of Theorem 5.3 in [23]. The setting in this paper is different from [23] in that there is a budget constraint in addition to the ROS constraint. This makes the analysis of bidders' spend different.

In addition, including the budget constraint means that the optimal assignment may be randomized while the setting in [23] always has an optimal assignment which is deterministic. Consider the following example with two bidders and one query: $B_1 = B_2 = 1, v_{11} = v_{21} = 2$. The optimal assignment is to assign the query to each bidder with a 0.5 probability, which achieves a total expected liquid welfare of 2, but any deterministic assignment only has a total liquid welfare of 1.

Fix any equilibrium EQ with two bidders, we use $V_i$ instead of $V_i^{\mathrm{EQ}}$ to denote the value bidder $i$ obtains in the equilibrium. We split all the advertisers into two sets:

$$A_B = \{i \mid B_i \leq V_i\} \text{ and } A_{\bar{B}} = A \setminus A_B.$$

For $i \in A_B$, $\mathrm{LW}(i) = B_i$, which is the maximum liquid welfare bidder $i$ can obtain. Thus, we have:

$$\mathrm{LW}(A_B) \geq \mathrm{OPT}(A_B). \tag{15}$$

Fix $i \in A_{\bar{B}}$. We now consider three cases depending on $\pi_{i,j}$ relative to $\pi_{i,j}^*$.

*Case 1:* $\pi_{i,j}^* \leq \pi_{i,j}$. In this case, we have $v_{i,j} \cdot \pi_{i,j}^* \leq v_{i,j} \cdot \pi_{i,j}$. We let $Q_{i,1} = \{j \in Q : \pi_{i,j}^* \leq \pi_{i,j}\}$.

*Case 2:* $\pi_{i,j}^* > 0$ *and* $\pi_{i,j} = 0$. By Lemma B.1, $\mathrm{SPEND}(j) = b_{\bar{i},j} \geq \alpha v_{i,j} \geq \alpha \cdot \pi_{i,j}^* \cdot v_{i,j}$. We let $Q_{i,2} = \{j \in Q : \pi_{i,j}^* > 0, \pi_{i,j} = 0\}$.

*Case 3:* $0 < \pi_{i,j} < \pi_{i,j}^* < 1$. Fix a query $j$ and let $\beta_j = b_{i,j}/b_{\bar{i},j} \in (1/\alpha, \alpha)$. Define $m_\beta = \frac{1}{2} \ln \left(1 + \frac{\ln \beta}{\ln \alpha}\right)$. We have that $\pi_{i,j} = m_\beta$. Next, by Lemma B.2, we have $b_{i,j} \geq \frac{v_{i,j}}{1+\ln(\alpha)+\ln(\beta_j)}$. In particular, we have

$\mathrm{SPEND}(j)$

$= m_\beta \cdot b_{i,j} + (1 - m_\beta) \cdot b_{\bar{i},j}$

$\geq m_\beta \cdot \dfrac{v_{i,j}}{1 + \ln \alpha + \ln \beta} + (1 - m_\beta) \cdot \dfrac{v_{i,j}}{\beta \cdot (1 + \ln \alpha + \ln \beta)}$

$= v_{i,j} \cdot s_\beta \geq \pi_{i,j}^* \cdot v_{i,j} \cdot s_\beta.$

where $s_\beta = \dfrac{1 + \frac{\ln \beta}{\ln \alpha}}{2(1 + \ln \alpha + \ln \beta)} + \dfrac{1 - \frac{\ln \beta}{\ln \alpha}}{2\beta(1 + \ln \alpha + \ln \beta)}$. For a fixed $\beta$, we define the set $Q_{i,3}^\beta = \{j \in Q : b_{i,j}/b_{\bar{i},j} = \beta\}$.

Since the number of queries $Q$ is finite, there exists a finite list $\beta_1, \ldots, \beta_k$ such that

$$Q = \bigcup_{i \in [2]} \left(Q_{i,1} \cup Q_{i,2} \cup (\cup_{\ell=1}^k Q_{i,3}^{\beta_\ell})\right).$$

We now lower bound the liquid welfare in several ways. Later, we use these lower bounds to show that some linear combination of them give an upper bound on the optimal liquid welfare which yields our desired approximation ratio.

First, we note that the total liquid welfare is an upper bound on the total spend. Thus, we have

$$\mathrm{LW} \geq \sum_{j \in Q} \mathrm{SPEND}(j) \geq \sum_{i \in A_{\bar{B}}} \sum_{j \in Q_{i,2}} \mathrm{SPEND}(j)$$

$$+ \sum_{i \in A_{\bar{B}}} \sum_{\ell=1}^k \sum_{j \in Q_{i,3}^{\beta_\ell}} \mathrm{SPEND}(j) \tag{16}$$

$$\geq \sum_{i \in A_{\bar{B}}} \sum_{j \in Q_{i,2}} \alpha \cdot \pi_{i,j}^* \cdot v_{i,j} + \sum_{i \in A_{\bar{B}}} \sum_{\ell=1}^k \sum_{j \in Q_{i,3}^{\beta_\ell}} s_{\beta_\ell} \cdot \pi_{i,j}^* \cdot v_{i,j} \tag{17}$$

$$= \sum_{i \in A_{\bar{B}}} \sum_{j \in Q_{i,2}} \alpha \cdot v_{i,j}^* + \sum_{i \in A_{\bar{B}}} \sum_{\ell=1}^k \sum_{j \in Q_{i,3}^{\beta_\ell}} s_{\beta_\ell} \cdot v_{i,j}^*. \tag{18}$$

In Eq. (16), we made use of the fact that the sets $\{Q_{i,2}\}_{i \in A_{\bar{B}}} \cup \{Q_{i,3}^{\beta_\ell}\}_{i \in A_{\bar{B}}, \ell \in [k]}$ are all pairwise disjoint (Claim B.3) and thus, the RHS is a valid lower bound on the total spend. For Eq. (17), we used the spend lower bounds from case 2 and case 3 described above.

*Claim B.3.* The sets $\{Q_{i,2}\}_{i \in A_{\bar{B}}}$, $\{Q_{i,3}^{\beta_\ell}\}_{i \in A_{\bar{B}}, \ell \in [k]}$ are all pairwise disjoint.

PROOF. For fixed $i \in A_{\bar{B}}$, the sets $\{Q_{i,2}\}, \{Q_{i,3}^{\beta_\ell}\}_{\ell \in [k]}$ are pairwise disjoint since $Q_{i,2}$ correspond to queries $j$ where $b_{i,j} \leq b_{\bar{i},j}/\alpha$ and $Q_{i,3}^\beta$ correspond to queries $j$ where $b_{i,j} = \beta b_{\bar{i},j}$. If $|A_{\bar{B}}| \leq 1$ then the claim is proved. Otherwise, we assume $A_{\bar{B}} = \{1, 2\}$. Let $Q_i = Q_{i,2} \cup \left(\cup_{\ell=1}^k Q_{i,3}^{\beta_\ell}\right)$. Note that if $j \in Q_i$ then $\pi_{i,j} < \pi_{i,j}^*$ and so it must be that $\pi_{\bar{i},j} > \pi_{\bar{i},j}^*$. Hence $j \notin Q_{\bar{i}}$. The claim is proved. □

Note that for $j \in Q_{i,1}$, we have $\pi_{i,j}v_{i,j} \geq \pi_{i,j}^*v_{i,j}$ (by definition of $Q_{1,i}$) and for $j \in Q_{i,3}^\beta$, we have $\pi_{i,j}v_{i,j} = m_\beta v_{i,j} \geq m_\beta v_{i,j}^*$. Thus, we have

$$\text{LW}(A_{\bar{B}}) \geq \sum_{i \in A_{\bar{B}}} \sum_{j \in Q_{i,1}} \tilde{v}_{i,j} + \sum_{i \in A_{\bar{B}}} \sum_{\ell=1}^k \sum_{j \in Q_{i,3}^{\beta_\ell}} \tilde{v}_{i,j}$$

$$\geq \sum_{i \in A_{\bar{B}}} \sum_{j \in Q_{i,1}} v_{i,j}^* + \sum_{i \in A_{\bar{B}}} \sum_{\ell=1}^k \sum_{j \in Q_{i,3}^{\beta_\ell}} m_{\beta_\ell} \cdot v_{i,j}^*. \quad (19)$$

Let $\text{OPT}(A_{\bar{B}})$ be the total optimal liquid welfare for all bidders in $A_{\bar{B}}$. In other words,

$$\text{OPT}(A_{\bar{B}}) = \sum_{i \in A_{\bar{B}}} \sum_{j \in Q_{i,1} \cup Q_{i,2} \cup (\cup_{\ell=1}^k Q_{i,3}^{\beta_\ell})} v_{i,j}^* \quad (20)$$

Combining Eq. (18), Eq. (19), and Eq. (20), if $\gamma, \eta \geq 0$, we have that

$$\eta\text{LW} + \gamma\text{LW}(A_{\bar{B}})$$

$$\geq \sum_{i \in A_{\bar{B}}} \sum_{j \in Q_{i,1}} \gamma v_{i,j}^* + \sum_{i \in A_{\bar{B}}} \sum_{j \in Q_{i,2}} \alpha\eta v_{i,j}^*$$

$$+ \sum_{i \in A_{\bar{B}}} \sum_{\ell=1}^k \sum_{j \in Q_{i,3}^{\beta_\ell}} (\eta m_{\beta_\ell} + \gamma s_{\beta_\ell}) v_{i,j}^*$$

$$\geq \min\left\{\gamma, \alpha\eta, \min_{\beta \in [1/\alpha, \alpha]} \eta m_\beta + \gamma s_\beta\right\} \cdot \sum_{i \in A_{\bar{B}}} \sum_{j \in Q} v_{i,j}^*$$

$$= \min\left\{\gamma, \alpha\eta, \min_{\beta \in [1/\alpha, \alpha]} \eta m_\beta + \gamma s_\beta\right\} \cdot \text{OPT}(A_{\bar{B}}).$$

[23] show that for $\alpha = 1.4$, if $\eta = 0.44$ and $\gamma = 0.56$ then $\min\left\{\gamma, \alpha\eta, \min_{\beta \in [1/\alpha, \alpha]} \eta m_\beta + \gamma s_\beta\right\} \geq 1/1.8$. Combining with Eq. (15), we conclude that

$$\text{LW} = (\eta + \gamma) \cdot \text{LW} = (\eta\text{LW} + \gamma\text{LW}(A_{\bar{B}})) + \gamma\text{LW}(A_B)$$

$$\geq \frac{1}{1.8} \cdot \text{OPT}(A_{\bar{B}}) + \gamma\text{OPT}(A_B) \geq \frac{1}{1.8}\text{OPT},$$

since $\gamma = 0.56 \geq \frac{1}{1.8}$. □

## B.2 Proof of Theorem 4.2

PROOF OF THEOREM 4.2. The proof is similar to Theorem 4.1. Case 1 and Case 2 are the same as in Theorem 4.1. For Case 3, we have the following lemma instead of Lemma B.2:

LEMMA B.4. *Suppose the bidders are assumed to bid uniformly. Fix bidder $i$ and query $j$. For any set of undominated bids, if $\pi_{i,j} \in (0, 1)$ and $V_i < B_i$ then $b_{i,j} = v_{i,j}$.*

PROOF. We denote bidder $i$'s uniform bid multiplier as $m_i$, i.e. for every query $j \in Q, b_{i,j} = m_i \cdot v_{i,j}$. Let $\pi$ be the allocation in equilibrium EQ. It is easy to see that $m_i \leq 1$ when $V_i > 0$. Assume the opposite that $m_i > 1$, then we have $b_{i,j} > v_{i,j}$. If there exists $\pi_{i,j} > 0$, then $\sum_{j \in Q} \pi_{i,j} \cdot b_{i,j} > \sum_{j \in Q} \pi_{i,j} \cdot v_{i,j}$, i.e., the ROS constraint is violated.

Next we show that $m_i \geq 1$. Assume the opposite that $m_i < 1$. Let $\text{COST}(m)$ be the cost function of bidder $i$'s uniform multiplier and $V(m)$ the the value function of bidder $i$'s uniform multiplier. Note that the ROS constraint is always satisfied as long as $m_i \leq 1$. So the only reason why $m_i < 1$ would be that if bidder $i$ deviates to $m = 1$, they may violate their budget constraint i.e. $B_i < \text{COST}(1) = V(1)$. In the equilibrium EQ, we have $m_i < 1$, and $\text{COST}(m_i) < V(m_i) < B_i$ where the last inequality is by the hypothesis of this lemma. Because $\text{COST}(m)$ and $V(m)$ are both monotone non-decreasing functions, there must exist $m' \in (m_i, 1)$ such that $V(m') > V(m_i)$ and $\text{COST}(m') < B_i$. This contradicts the fact that bidder $i$ is in an equilibrium. We conclude that $m_i = 1$. □

*Case 3:* $0 < \pi_{i,j} < \pi_{i,j}^* < 1$. Fix a query $j$ and let $\beta_j = b_{i,j}/b_{\bar{i},j} \in (1/\alpha, \alpha)$. Define

$$m_\beta = \frac{1}{2}\ln\left(1 + \frac{\ln \beta}{\ln \alpha}\right).$$

We have that $\pi_{i,j} = m_\beta$. Next, by Lemma B.4, we have $b_{i,j} \geq v_{i,j}$. In particular, we have

$$\text{SPEND}(j) = m_\beta \cdot b_{i,j} + (1 - m_\beta) \cdot b_{\bar{i},j}$$

$$\geq m_\beta \cdot v_{i,j} + (1 - m_\beta) \cdot \frac{v_{i,j}}{\beta}$$

$$= v_{i,j} \cdot s_\beta$$

$$\geq \pi_{i,j}^* \cdot v_{i,j} \cdot s_\beta,$$

where $s_\beta = m_\beta + \frac{1 - m_\beta}{\beta}$.

Proceeding as in the proof of Theorem 4.2, we conclude that PoA $\leq 1.5$ when $\alpha = 7.63$ and $\eta = 0.33$. □

## C MISSING PROOFS FOR SECTION 5

### C.1 Proof of Lemma 5.2

PROOF OF LEMMA 5.2. Since the query $j$ is fixed, we drop the subscript $j$. We also set $i = 1$. First observe that the condition

Pr $[i \text{ wins query } j] \leq \eta$ is equivalent to $\|b\|_\alpha^\alpha \geq b_1^\alpha/\eta$. Next, we prove a lower bound on $b_1$. Let $f(b) = \frac{(v-b) \cdot b^\alpha}{b^\alpha + \|b_{-1}\|_\alpha^\alpha}$ be the difference between the value that bidder 1 extracts on the query and their expected payment, if they bid $b$. Note that we must have that $f'(b_1) \leq 0$ otherwise bidder 1 can raise their bid by an infinitesimal amount to get more value while maintaining their ROS and budget constraint. Taking derivatives, we thus have that

$$f'(b) = -\frac{b^{\alpha-1} \cdot \left(b^{\alpha+1} + b\|b_{-1}\|_\alpha^\alpha \cdot (\alpha+1) - \|b_{-1}\|_\alpha^\alpha v\alpha\right)}{\|b\|_\alpha^{2\alpha}}.$$

We thus require that

$$0 \leq b^{\alpha+1} + b\|b_{-1}\|_\alpha^\alpha \cdot (\alpha + 1) - \|b_{-1}\|_\alpha^\alpha v\alpha$$

$$\leq b\|b_{-1}\|_\alpha^\alpha \cdot \left( \frac{1}{1/\eta - 1} + \alpha + 1 \right) - \|b_{-1}\|_\alpha^\alpha v\alpha,$$

whence, $b \geq v\alpha \cdot \frac{1/\eta - 1}{\alpha/\eta - \alpha + 1/\eta}$. Now, we lower bound the spend. We have that

$$\textsc{Spend}(j) = \sum_{i=1}^n \frac{b_i^{\alpha+1}}{\|b\|_\alpha^\alpha} \geq \frac{\|b\|_\alpha}{n^{1/\alpha}} \geq \frac{b_1}{(n\eta)^{1/\alpha}}$$

$$\geq v \cdot \frac{\alpha/\eta - \alpha}{(n\eta)^{1/\alpha}(\alpha/\eta - \alpha + 1/\eta)}.$$

Multiplying the numerator and denominator by $\eta$ gives the claim. $\square$

