# OpenReview forum: "Efficiency of Non-Truthful Auctions in Auto-bidding with Budget Constraints"
_ACM.org/TheWebConf/2024/Conference — TheWebConf24 Oral_

### Official Review · Reviewer_cZGF · 2023-11-17

**Novelty:** 5
**Technical Quality:** 4

**Review:**

This paper studies the efficiency of First Price Auction (FPA) in auto-bidding with return on spend (ROS) and budget constraints. The efficiency is measured by the price of anarchy (PoA), which is the worst ratio between the liquid welfare of optimal fractional allocation and any equilibrium. This paper first shows that under both ROS and budget constraints, the PoA of FPA is n, which is tight. With a natural assumption that each bidder’s value for any query does not exceed her budget, the tight PoA is 2, which also matches the tight result without budget constraints. This paper also introduces the Integral Price of Anarchy (I-PoA) which is the worst ratio between the liquid welfare of optimal integral allocation and any equilibrium. They further show that the I-PoA of FPA is 2, which is also tight. Besides FPA, this paper proposes two randomized mechanisms called randomized FPA (rFPA) and quasi-proportional FPA and shows that the PoA of rFPA is 1.8, and the PoA of quasi-proportional FPA is 2.

Strengths:
1)	The auto-bidding problem is a central problem in the Web Conference since it has rich applications related to the web internet and economics, e.g., online advertising.
2)	This paper studies a natural extension of auto-bidding, by considering both ROS and budget constraints, which fills the gaps in previous research. At the same time, this paper also draws a near-complete picture for the efficiency of FPA, under these two constraints.

Weaknesses:
1)	The technical contribution of the article is not very strong. While the results are almost complete, most of them are based on similar analyses. They can be regarded as a natural extension of each other, e.g., Theorem 3.3, Theorem 3.6, and Theorem 3.8.
2)	The structure of the paper can be improved. For example, Section 3 covers most of the results of this paper: including the upper and lower bounds of PoA of FPA, the upper and lower bounds of I-PoA of FPA, and the upper and lower bounds of PoA under the small value assumption. However, all these results are placed in the same subsection. It would be better if they were separated in 2 to 3 subsections.

**Questions:**

Regarding the quasi-proportional FPA, in my understanding, when n approaches infinity, this mechanism essentially does the same thing as the (orginal) FPA? (Allocate as much as possible to the largest bidder). But their PoA is very different, one is 2 and the other is n. Can you explain why?

**Reviewer Confidence:**

3: The reviewer is confident but not certain that the evaluation is correct

**Scope:**

4: The work is relevant to the Web and to the track, and is of broad interest to the community

---

### Official Review · Reviewer_txby · 2023-11-19

**Novelty:** 6
**Technical Quality:** 6

**Review:**

### Summary

The paper studies the performance of first-price auctions (FPA) and variants in the autobidding world, with both ROS constraints and budget constraints.  The authors present fairly comprehensive results, including matching upper and lower bounds in natural several settings, as well as upper bounds for randomized and "smoothed" versions of FPA.


### Strengths

The problem studied is of theoretical and practical importance.  The focus on budget constraints is, to my knowledge, a novel perspective.  To this end, the paper presents a strong conceptual message: budget constraints do make things significantly different for FPA.  The proofs contain ideas that might be useful for future research in autobidding with budget constraints.


### Weaknesses

The paper is a bit specific in that it talks almost exclusively about FPA and variants.  Another gap that I'd like to see filled is the (non)existence of equilibria, which the authors also mention as a future direction.

**Questions:**

Around line 103, definition of PoA: defined this way the PoA should be at most 1?

Line 157, "in a couple setting": "a couple of settings"?

Lines 390 and 392: extra "("

Line 427, "OPT": I know what you mean but technically OPT is a real number...

Line 449, "when a Nash equilibrium exists": is there anything you can say about its existence?

FPA + uniform bidding: the authors say the source of inefficiency is the "all or nothing" phenomenon, which makes sense.  So I'm curious what would happen in a smoothed world.  For example, if we take the worst-case instance and add a tiny noise to every value, is it still possible to find an instance (before adding noise) that forces a PoA of almost n?  Relatedly, is it possible that rFPA gets a much better PoA than n with n > 2 bidders?

**Reviewer Confidence:**

3: The reviewer is confident but not certain that the evaluation is correct

**Scope:**

4: The work is relevant to the Web and to the track, and is of broad interest to the community

---

### Official Review · Reviewer_a1B5 · 2023-11-21

**Novelty:** 5
**Technical Quality:** 5

**Review:**

The authors study the efficiency of non-truthful auctions under auto-bidding and budget constraints. In particular, in the studied model, each advertiser has a value for each query, and their goal is to maximize their total value so that this value does not exceed their total spend. Then, the auto-bidding agents try to solve this optimization problem on behalf of the advertisers. Under this setting, the authors consider the first price auction (FPA), which is known to be non-truthful, and prove the following:

1) The FPA is optimal among all deterministic mechanisms in this setting. Specifically, its price of anarchy (PoA) is n (where n is the number of advertisers), and this is a bound for any deterministic mechanism. For the special case where the value of a bidder for any query does not exceed their total budget, the prove that the PoA is 2.
2) The authors consider 2 randomized versions of the FPA, and (among others) they show that the PoA can be improved to 2, bypassing thus the impossibility results that hold for deterministic mechanisms.
3) Finally, the authors consider bidders that bid uniformly, and they show that this affects negatively the performance (in terms of efficiency) of deterministic mechanisms, while it is beneficial for randomized mechanisms.

Strengths

1) The paper studies an interesting and well-motivated problem. It is in general well-written, and has a clear focus.
2) The authors provide a nice collection of results, give a clear picture of the performance of the celebrated FPA in this setting, and this picture is also complete as the paper considers both the case of deterministic and randomized mechanisms
3) The paper technically is not trivial, and as far as I checked is sound and correct.

Weaknesses

I do not have any major complaints apart from the fact that, although as I said the paper is well-written, it is not always easy to follow. A revision on the introductory sections so that the model is more clear would be appreciated.

Overall, I would say that this is a nice paper, with an interesting set of results. It is not always easy to follow but regardless, I believe that it  is a good match for the conference.

**Questions:**

None.

**Reviewer Confidence:**

3: The reviewer is confident but not certain that the evaluation is correct

**Scope:**

3: The work is somewhat relevant to the Web and to the track, and is of narrow interest to a sub-community

---

### Official Review · Reviewer_eT6T · 2023-11-21

**Novelty:** 5
**Technical Quality:** 5

**Review:**

The authors consider the problem of designing auctions for value-maximizing bidders with both return on spend and budget constraints (i.e., an auto-bidding context).  An auctioneer aims to maximize the social welfare whereas the bidders attempt to maximize their own individual value subject to their spend not exceeding a budget constraint or a return on spend constraint.  In particular, the authors examine the price of anarchy (i.e., ratio between the optimal achievable liquid welfare over the liquid welfare at any equilibrium) of the (deterministic) first-price auction and the randomized first-price auction and quasi-proportional first-price auction.

First, the authors show that the first-price auction obtains a price of anarchy of $n$ (which is optimal among deterministic mechanisms) and this bound improves to $2$ with the additional assumption that bidders do not have any value for any query exceeding their budget.  Secondly, they demonstrate how the use of randomization can lead to improved guarantees by showing that the randomized first price auction obtains a price of anarchy no greater than $1.8$ when there exists only two bidders and that the quasi-proportional first-price auction achieves a price of anarchy $2$ without any additional assumptions.

On the whole, this paper makes a substantial contribution to the literature on the price of anarchy of different auction formats in the auto-bidding context.  The first-price auction and its variants are well-known and practically-favored mechanisms for advertising auctions and, thus, better understanding their welfare guarantees in settings in which bidders have both return-on-spend and budget constraints is natural.  Finally, the results are neatly presented and the paper is largely well-written.  While it is not completely surprising that many of the price of anarchy results previously shown in the presence of only return-on-spend constraints when social welfare is the objective carry over to the setting with added budget constraints when liquid welfare is the objective, the proofs are non-trivial and the authors provide a nice separation between the two settings in the case of uniform bidding.  As such, while the results are somewhat narrow, I believe they are likely to be of interest to the portion of the community interested in auto-bidding models.

While I outline some smaller presentational issues below, there is one larger issue I noticed with the presentation.  In particular, I am slightly confused by the proof of Section 5.  You say that the price of anarchy converges to $1/\eta + 1$ and then conclude that the price of anarchy is $2$.  However, the assumption in the statement of Lemma 5.3 is only that $\eta \in (0,1)$.  The price of anarchy tends to $2$, then, if you are allowed to choose $\eta$ but $\infty$ if you are not allowed to choose $\eta$.  When you state that “$\eta$ is arbitrary”, it reads like an adversary would get to set it.  This proof and surrounding argument should be made much clearer to show that the price of anarchy indeed is $2$.


Line 367: “total expected is no more” -> “total expected spend is no more”

Line 390:  There is an extra parenthesis in the definition of OPT

Line 487: “an Equilibrium” -> “an equilibrium”

Line 723: I believe you want the inequality to use SPEND$(j’)$.

Line 694 and 742:  In the statement of the lemma (or in the proof itself at line 742) you probably want to point out that the inequality regarding the $\pi^*$ summation being less than $1$ is by assumption.

[After rebuttal]  The authors have, in my view, adequately addressed my question and I appreciate that they will clarify the proof in future revisions.

**Questions:**

Can you clarify the proof of Section 5, particularly regarding the point about $\eta$ being arbitrary referenced in the review?

**Reviewer Confidence:**

3: The reviewer is confident but not certain that the evaluation is correct

**Scope:**

4: The work is relevant to the Web and to the track, and is of broad interest to the community

---

### Official Review · Reviewer_CnoD · 2023-12-01

**Novelty:** 5
**Technical Quality:** 5

**Review:**

This is a very well-written paper with very strong results. Its contributions are as follows:

1. It shows that non-truthfulness in bidding has no extra utility in the deterministic auction setting. This is shown by achieving a price of anarchy that matches that achieved by truthful bidders (in the deterministic setting) in the case of two bidders.

2. In the randomized setting, it shows that nontruthfulness does provide a benefit by showing a PoA of 1.8 (in contrast with that of 1.9 as achieved by the truthful mechanism) in the case of two bidders.

3. Finally, the paper shows that no auction (even randomized, non-truthful) can
improve upon a PoA bound of 2 as the number of advertisers grow to infinity.

In all these settings, the paper considers the bidder to have a value-maximizing objective and budget and ROS constraints. This is in contrast to much of prior work that assumes only the ROS constraint.

The proofs are all simple but very clever, and I think the results are strong.

Even though this is an "auctions" paper, I think it would be good to cite some of the related "algorithms" papers dealing with ROS + budget constraints, e.g.:

1. Online Bidding in Repeated Non-Truthful Auctions under Budget and ROI constraints by Castiglioni, Celli, Kroer


2. Online Bidding Algorithms for Return-on-Spend Constrained Advertisers by Feng, Padmanabhan, Wang

I know there are a few more published in last year's NeurIPS and ICML, I think those could be included too.

**Questions:**

N/A

**Reviewer Confidence:**

3: The reviewer is confident but not certain that the evaluation is correct

**Scope:**

4: The work is relevant to the Web and to the track, and is of broad interest to the community

---

### Decision · Program_Chairs · 2024-01-22

**Decision:**

Accept (Oral)

**Comment:**

I will spare the authors another summary of their work, given the detailed reviews prepared by several of the PC members -- I thank them for their efforts in evaluating this work, both in the original reviews and in the subsequent rebuttal period.

 Given that, across the 4 reviewers, there is ample support for this paper (and no real opposition), I recommend acceptance.